# Interval-Valued Cores and Interval-Valued Dominance Cores of Cooperative Games Endowed with Interval-Valued Payoffs

**Hsien-Chung Wu** 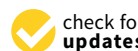

Department of Mathematics, National Kaohsiung Normal University, Kaohsiung 802, Taiwan;
hcwu@nknucc.nknu.edu.tw

**Abstract:** Cooperative games endowed with interval-valued payoffs are studied in this paper. Based on the interval-valued payoff and the different types of orderings, we can propose many types of so-called interval-valued cores and interval-valued dominance cores. The main issue of this paper is to establish the equalities of different types of interval-valued cores and interval-valued dominance cores under a mild assumption. Without considering the individual rationality, we also establish the equalities of different types of interval-valued pre-cores and interval-valued dominance pre-cores without any extra assumptions.

**Keywords:** cores; dominance cores; Hukuhara difference; interval-valued payoffs; null set

## 1. Introduction

The game theory in economics comes from the pioneering work by von Neumann and Morgenstern [1]. The game theory mainly concerns the behavior of players whose decisions affect each other. The topic of cooperative games can also be regarded as games in coalitional form. Nash [2] defined the concept of a general two-person cooperative game and a solution concept of such games. The cooperation means that the players have complete freedom of communication and comprehensive information on the structure of the game. Young [3] studied the monotonic solutions of cooperative games. The principle of monotonicity for cooperative games states that if a game changes such that some player's contribution to all coalitions increases or stays the same, then the player's allocation should not decrease. The well-known Shapley value is a unique symmetric and efficient solution concept that is also monotonic.

In a cooperative game, the payoff of a coalition is assumed to be a real number that can also represent the worth under this coalition. Owing to unexpected situations and fluctuation in the real market, the payoff cannot be measured precisely. In other words, the payoff is uncertain in the real world.

Based on the probability theory, the stochastic payoffs were studied by Fernández et al. [4], Granot [5] and Suijs et al. [6], where the payoffs were assumed to be random variables. The other types of stochastic games were also proposed by Chakrabarti [7], Alvarez-Mena and Hernández-Lerma [8], Dshalalow and Huang [9], Dekel et al. [10] and Guo and Yang [11]. A general model of cooperative games whose characteristic function takes values in a partially ordered linear space has also been studied by Puerto et al. [12].

Another concept of uncertain payoff is to consider the interval-valued payoff in which the uncertain worth of a given coalition is assumed to be located in a bounded closed interval instead of assuming it to be a random variable. The advantage with considering interval-valued payoffs is that the determination of uncertainty is easily performed by simply determining the lower and

upper end-points. However, the determination of uncertainty for stochastic payoffs by providing the probability distribution functions is not an easy task.

Applying the interval analysis (ref. Moore [13]) to the conventional cooperative game establishes the so-called interval-valued cooperative games by referring to Branzei et al. [14], Alparslan Gok et al. [15] and Mallozzi et al. [16]. The cooperative games endowed with the interval-valued payoffs has also been investigated in Alparslan Gök et al. [17] and the references therein. The inclusion relationship between the interval-valued core and interval-valued dominance core has been established in Alparslan Gök et al. [17] by showing that the interval-valued core is contained in the interval-valued dominance core. Since the equalities of the interval-valued core and interval-valued dominance core were not established in Alparslan Gök et al. [17], in this paper, we shall establish the equalities of the interval-valued core and interval-valued dominance core by introducing different types of orderings.

Branzei et al. [18] considered the interval-valued bankruptcy games that arose from bankruptcy situations with interval-valued claims in which two interval-valued Shapley-like values are proposed and the interrelations using the interval arithmetic operations (ref. Moore [13]) are studied. Mallozzi et al. [16] introduced a solution concept of core-like cooperative games in which the fuzzy intervals were taken into account. Also, a necessary condition was provided to assure the non-emptiness of core-like games. The more detailed fuzzy cooperative games can refer to Mares [19]. On the other hand, Han et al. [20] proposed the interval-valued core and the interval Shapley-like value of cooperative games by defining a new order relation of intervals. Alparslan Gok et al. [21] considered the Weber set and the Shapley value for interval-valued cooperative games and established their relations with the interval-valued core of convex interval-valued cooperative games. Branzei et al. [22] and Alparslan Gok et al. [17] also studied the interval-valued core, the interval-valued dominance core, the interval-valued square dominance core and the interval-valued stable sets of cooperative games under interval uncertainty. Li et al. [23–25] proposed several concepts of interval-valued solutions such as the interval-valued Shapley value, the interval-valued solidarity value and the interval-valued Banzhaf value in which an effective nonlinear programming method for computing interval-valued cores was established. Meng et al. [26] proposed a generalized form of fuzzy games with interval characteristic functions in which the interval Shapley function and interval population monotonic allocation function were studied.

A bounded closed interval subtracting itself cannot be a zero element. Therefore the additive inverse element of a bounded closed interval does not exist. In this paper, we introduce the concept of a null set that can be used to define the orderings and almost identical concept in the space consisting of all bounded and closed intervals. We also introduce the concept of Hukuhara difference for the bounded closed intervals, which is used to propose the different types of cores and dominance cores. The argument for studying the relationships between the interval-valued core and interval-valued dominance core in this paper is completely different from that of Alparslan Gök et al. [17], since the concepts of null set and Hukuhara difference are invoked. Especially, based on these settings, we can establish the equalities of the interval-valued core and interval-valued dominance core, which was not studied in Alparslan Gök et al. [17].

In Section 2, we propose the concept of a null set and present the interesting properties of bounded closed intervals in $\mathbb{R}$ that will be used for the further investigation. We also introduce many types of orderings that will be used to study the interval-valued cores and interval-valued dominance cores. In Section 3, we introduce the concept of a cooperative game endowed with the interval-valued payoffs. In Sections 4 and 5, based on the different orderings introduced in Section 2, we propose many types of cores and dominance cores. On the other hand, using the Hukuhara difference, we also introduce the so-called H-cores and dominance H-cores. In Section 6, we study the relationships between cores and dominance cores. Without considering the individual rationality, in Section 7, we also study the relationships between pre-cores and dominance pre-cores.

## 2. Intervals in $\mathbb{R}$

We write $[a, b]$ to denote a bounded closed interval in $\mathbb{R}$, where $a$ is the left-endpoint and $b$ is the right-endpoint. The *center* of $[a, b]$ is defined by the average $c \equiv (a + b)/2$ of the left-endpoint and right-endpoint. The bounded closed interval $[a, b]$ can also be regarded as an "approximated real number $c$" with *symmetric uncertainty* $u \equiv b - c = c - a$, where $u$ is the half-width of the bounded closed interval $[a, b]$. We denote by $\mathcal{I}$ the collection of all bounded closed intervals in $\mathbb{R}$. For convenience, for any $A \in \mathcal{I}$, we write $A = [a^L, a^U]$.

Let $\odot$ denote any of the four basic arithmetic operations $\oplus, \ominus, \otimes, \oslash$ between two bounded closed intervals $A = [a^L, a^U]$ and $B = [b^L, b^U]$. We define

$$A \odot B = \{c : c = a + b \text{ for all } a \in A \text{ and } b \in B\}.$$

Then we have the following operations:

$$A \oplus B = \left[ a^L + b^L, a^U + b^U \right]$$
$$A \ominus B = \left[ a^L - b^U, a^U - b^L \right],$$
$$A \otimes B = \left[ \min \left\{ a^L b^L, a^L b^U, a^U b^L, a^U b^U \right\}, \max \left\{ a^L b^L, a^L b^U, a^U b^L, a^U b^U \right\} \right],$$
$$A \oslash B = \left[ \min \left\{ a^L/b^L, a^L/b^U, a^U/b^L, a^U/b^U \right\}, \max \left\{ a^L/b^L, a^L/b^U, a^U/b^L, a^U/b^U \right\} \right] \text{ for } 0 \notin B.$$

Since $A \odot B$ is a bounded closed interval, we also write $(A \odot B)^L$ and $(A \odot B)^U$ to denote the left-endpoint and right-endpoint of $A \odot B$, respectively.

In particular, we have

$$A \ominus A = \left[ a^L - a^U, a^U - a^L \right] = \left[ -\left( a^U - a^L \right), a^U - a^L \right],$$

which says that each $A \ominus A$ is an "approximated zero" with symmetric uncertainty $a^U - a^L$. Therefore, we say that $A \ominus A$ is an *interval zero* . The *zero interval* is defined to be $[0, 0]$.

Let $\Omega = \{A \ominus A : A \in \mathcal{I}\}$ that collects all interval zeros. Equivalently, $\omega \in \Omega$ if and only if $\omega^U \geq 0$ and $\omega^L = -\omega^U$, i.e.,

$$\omega = \left[ \omega^L, \omega^U \right] = \left[ -\omega^U, \omega^U \right],$$

where the bounded closed interval $\omega$ is an "approximated zero" with symmetric uncertainty $\omega^U$. We also call $\Omega$ as the *null set* in $\mathcal{I}$. It is also clear that the zero interval $[0, 0]$ is in the null set $\Omega$.

**Remark 1.** *It is not hard to check that the null set $\Omega$ is closed under the interval addition. In other words, for any $\omega_1, \omega_2 \in \Omega$, we have $\omega_1 \oplus \omega_2 \in \Omega$.*

Let $A$ and $B$ be two bounded closed intervals. We say that $A$ and $B$ are *almost identical* if and only if the centers of $A$ and $B$ are equal. Suppose that their common centers are $\kappa \in \mathbb{R}$. Then $A$ and $B$ can both be regarded as the "approximated real number $\kappa$". The only difference is the uncertainty. In this case, if $A \subset B$ then we say that $B$ is more uncertain than $A$.

For any bounded closed interval $A = [a^L, a^U]$ and any $\omega = [\omega^L, \omega^U]$, we see that $A$ and $A \oplus \omega$ are almost identical in the sense of $A \oplus \omega$ being more uncertain than $A$. Therefore we can define the almost identical concept in $\mathcal{I}$ below.

**Definition 1.** *Given any two bounded closed intervals $A$ and $B$, we define $A =_\Omega B$ if and only if there exist $\omega_1, \omega_2 \in \Omega$ such that $A \oplus \omega_1 = B \oplus \omega_2$.*

**Remark 2.** *Suppose that $A =_\Omega B$. Since $[0,0] \in \Omega$, it can happen that one of the following situations is satisfied:*

- *$A = B$ when $\omega_1 = \omega_2 = [0,0]$;*
- *there exists $\omega \in \Omega$ such that $B \oplus \omega = A$ when $\omega_1 = [0,0]$ and $\omega_2 = \omega$;*
- *there exists $\omega \in \Omega$ such that $A \oplus \omega = B$ when $\omega_2 = [0,0]$ and $\omega_1 = \omega$.*

It is clear that $A \oplus (B \ominus B) =_\Omega A$. More interpretation regarding the equality $A =_\Omega B$ is presented below. We first observe that the center of $A \oplus \omega_1$ is $(a^L + a^U)/2$ and the center of $B \oplus \omega_2$ is $(b^L + b^U)/2$. This says that if $A =_\Omega B$ then the centers of $A$ and $B$ are all identical. Therefore we may say that $A =_\Omega B$ means $A$ and $B$ being identical with different symmetric uncertainties.

**Proposition 1.** *Let $A$ and $B$ be two bounded closed intervals.*

(i)    *If $C = A \ominus B$, then $C \oplus B =_\Omega A$.*
(ii)   *If $C = (A \oplus \omega_1) \ominus (B \oplus \omega_2)$ for some $\omega_1, \omega_2 \in \Omega$, then $C \oplus B =_\Omega A$.*
(iii) *Suppose that $\bar{A} =_\Omega A$ and $\bar{B} =_\Omega B$. Then we have the following properties.*

- *If $C = \bar{A} \ominus \bar{B}$, then $C \oplus B =_\Omega A$.*
- *If $C = (\bar{A} \oplus \omega_1) \ominus (\bar{B} \oplus \omega_2)$ for some $\omega_1, \omega_2 \in \Omega$, then $C \oplus B =_\Omega A$.*

**Proof.** It suffices to prove the second case of part (iii). Since $\bar{A} =_\Omega A$ and $\bar{B} =_\Omega B$, it follows that $\bar{A} \oplus \omega_3 = A \oplus \omega_4$ and $\bar{B} \oplus \omega_5 = B \oplus \omega_6$ for some $\omega^{(i)} \in \Omega$ for $i = 3, 4, 5, 6$. If $C = (\bar{A} \oplus \omega_1) \ominus (\bar{B} \oplus \omega_2)$, then, by adding $\bar{B} \oplus \omega_2$ on both sides, we have

$$C \oplus (\bar{B} \oplus \omega_2) = (\bar{A} \oplus \omega_1) \ominus (\bar{B} \oplus \omega_2) \oplus (\bar{B} \oplus \omega_2) = (\bar{A} \oplus \omega_1) \oplus \omega_7,$$

where

$$\omega_7 = (\bar{B} \oplus \omega_2) \ominus (\bar{B} \oplus \omega_2) \in \Omega.$$

By adding $\omega_3 \oplus \omega_5$ on both sides, we also have

$$C \oplus (\bar{B} \oplus \omega_5) \oplus \omega_2 \oplus \omega_3 = (\bar{A} \oplus \omega_3) \oplus \omega_5 \oplus \omega_1 \oplus \omega_7,$$

which implies

$$C \oplus (B \oplus \omega_6) \oplus \omega_2 \oplus \omega_3 = (A \oplus \omega_4) \oplus \omega_5 \oplus \omega_1 \oplus \omega_7.$$

Using Remark 1, we obtain

$$C \oplus B \oplus \omega_8 = A \oplus \omega_9,$$

where

$$\omega_8 = \omega_2 \oplus \omega_3 \oplus \omega_6 \in \Omega \text{ and } \omega_9 = \omega_1 \oplus \omega_4 \oplus \omega_5 \oplus \omega_7 \in \Omega.$$

This shows $C \oplus B =_\Omega A$, and the proof is complete. □

**Definition 2.** *Let $A$ and $B$ be two bounded closed intervals. We define three binary relations as follows:*

- *$A \preceq B$ if and only if $a^L \leq b^L$ and $a^U \leq b^U$;*
- *$A \preccurlyeq B$ if and only if $a^U \leq b^U$;*
- *$A \precsim B$ if and only if $a^U \leq b^L$.*

**Remark 3.** *It is clear that the above three binary relations are reflexive and transitive on $\mathcal{I}$; that is, they are partial orderings on $\mathcal{I}$. We also see that $A \precsim B$ implies $A \preceq B$, and that $A \preceq B$ implies $A \preccurlyeq B$. If the bounded closed intervals $A$ and $B$ are not degenerated, then $A = B$ cannot imply $A \precsim B$.*

**Definition 3.** *Let A and B be two bounded closed intervals. We define three binary relations as follows:*

- $A \prec B$ *if and only if $A \preceq B$ and $A \neq B$;*
- $A \sqsubset B$ *if and only if $a^U < b^U$;*
- $A \lessdot B$ *if and only if $a^U < b^L$.*

It is clear that
$$A \prec B \text{ if and only if } A \preceq B \text{ and } a^L < b^L \text{ or } a^U < b^U.$$

Therefore we also propose the following defintion.

**Definition 4.** *Let A and B be two bounded closed intervals. We define three binary relations as follows:*
$$A \lll B \text{ if and only if } A \preceq B \text{ and } a^L < b^L \text{ and } a^U < b^U.$$

**Remark 4.** *It is clear to see that $A \lessdot B$ implies $A \prec B$, that $A \lessdot B$ implies $A \sqsubset B$, and that $A \lll B$ implies $A \sqsubset B$.*

Since $\Omega$ can be regarded as a kind of "zero element" in $\mathcal{I}$, we also define the following binary relations.

**Definition 5.** *Given any two bounded closed intervals A and B, we define*
$$A \preceq_\Omega B \text{ if and only if } A \oplus \omega_1 \preceq B \oplus \omega_2$$

*for some $\omega_1, \omega_2 \in \Omega$. We can similarly define $A \precsim_\Omega B$ and $A \preccurlyeq_\Omega B$. The strict orderings*
$$A \prec_\Omega B, \quad A \sqsubset_\Omega B, \quad A \lessdot_\Omega B, \quad A \lll_\Omega B$$

*can also be similarly defined.*

Suppose that $A \preceq_\Omega B$. Since $[0,0] \in \Omega$, it can happen that one of the following situations is satisfied:

- $A \preceq B$ when $\omega_1 = \omega_2 = [0,0]$;
- there exists $\omega \in \Omega$ such that $A \preceq B \oplus \omega$ when $\omega_1 = [0,0]$ and $\omega_2 = \omega$;
- there exists $\omega \in \Omega$ such that $A \oplus \omega \preceq B$ when $\omega_2 = [0,0]$ and $\omega_1 = \omega$.

The same situations can also apply to $A \precsim_\Omega B$ and $A \preccurlyeq_\Omega B$.

The following different relations of transitivity will be used for discussing the inclusions of cores and dominance cores.

**Proposition 2.** *Let A, B and C be bounded closed intervals. We have the following properties.*

(i) *If $A \sqsubset B \precsim_\Omega C$ then $A \lessdot_\Omega C$, and if $A \sqsubset B \precsim C$ then $A \lessdot C$.*
(ii) *If $A \sqsubset B \preccurlyeq_\Omega C$ then $A \sqsubset_\Omega C$, and if $A \sqsubset B \preccurlyeq C$ then $A \sqsubset C$.*
(iii) *If $A \prec B \preceq_\Omega C$ then $A \prec_\Omega C$, and if $A \prec B \preceq C$ then $A \prec C$.*
(iv) *If $A \prec B \preccurlyeq_\Omega C$ then $A \sqsubset_\Omega C$, and if $A \prec B \preccurlyeq C$ then $A \sqsubset C$.*
(v) *If $A \prec B \precsim_\Omega C$ then $A \lessdot_\Omega C$, and if $A \prec B \precsim C$ then $A \lessdot C$.*
(vi) *if $A \lll B \preceq_\Omega C$ then $A \lll_\Omega C$, and if $A \lll B \preceq C$ then $A \lll C$;*

**Proof.** To prove part (i), by definition, there exist $\omega_1, \omega_2 \in \Omega$ such that $B \oplus \omega_1 \precsim C \oplus \omega_2$. We also have $a^U < b^U$. Therefore we obtain
$$a^U + \omega_1^U < b^U + \omega_1^U \leq c^L + \omega_2^L,$$

which says that $A \oplus \omega_1 \lessdot C \oplus \omega_2$. Without considering $\omega_1$ and $\omega_2$, we can similarly obtain that $A \sqsubset B \precsim C$ implies $A \lessdot C$. The other parts can be similarly obtained. This completes the proof. $\square$

**Remark 5.** *Based on Proposition 2, we can use Remarks 3 and 4 to obtain many other relations of transitivity as follows:*

- *Using (ii), if $A \sqsubset B \preceq_\Omega C$ then $A \sqsubset_\Omega C$, and if $A \sqsubset B \preceq C$ then $A \sqsubset C$.*
- *Using (i), if $A \lll B \precsim_\Omega C$ then $A \lessdot_\Omega C$, and if $A \lll B \precsim C$ then $A \lessdot C$;*
- *Using (ii), if $A \lll B \preccurlyeq_\Omega C$ then $A \sqsubset_\Omega C$, and if $A \lll B \preccurlyeq C$ then $A \sqsubset C$;*
- *Using (iii), if $A \lessdot B \preceq_\Omega C$ then $A \prec_\Omega C$, and if $A \lessdot B \preceq C$ then $A \prec C$;*
- *Using (iv), if $A \lessdot B \preccurlyeq_\Omega C$ then $A \sqsubset_\Omega C$, and if $A \lessdot B \preccurlyeq C$ then $A \sqsubset C$;*
- *Using (v), if $A \lessdot B \precsim_\Omega C$ then $A \lessdot_\Omega C$, and if $A \lessdot B \precsim C$ then $A \lessdot C$;*

## 3. Cooperative Games with Interval-Valued Payoffs

We consider a finite set $N = \{1, 2, \cdots, n\}$ of players. The subsets of $N$ are called the *coalitions*. We denote by $\mathcal{P}(N)$ the collection of all coalitions. Let $\mathfrak{v} : \mathcal{P}(N) \to \mathcal{I}$ be an interval-valued function defined on $\mathcal{P}(N)$.

Then $(N, \mathfrak{v})$ is called a *cooperative game with interval-valued payoff* if and only if $\mathfrak{v}(\varnothing) = [0, 0]$. Given a coalition $S$, owing to the unexpected situation, the payoff $\mathfrak{v}(S)$ regarding this coalition $S$ is assumed to be uncertain. In other words, the map $\mathfrak{v}$ assigns to each coalition a bounded closed interval telling what such a coalition can achieve in cooperation.

Let $(N, \mathfrak{v})$ be a cooperative game with interval-valued payoff. For any $S, T \subseteq N$ with $S \cap T = \varnothing$, we define three types of superadditivity as follows:

- We say that $(N, \mathfrak{v})$ is $\succeq$-*superadditive* if and only if

$$\mathfrak{v}(S \cup T) \succeq \mathfrak{v}(S) \oplus \mathfrak{v}(T);$$

- We say that $(N, \mathfrak{v})$ is $\succsim$-*superadditive* if and only if

$$\mathfrak{v}(S \cup T) \succsim \mathfrak{v}(S) \oplus \mathfrak{v}(T);$$

- We say that $(N, \mathfrak{v})$ is $\succcurlyeq$-*superadditive* if and only if

$$\mathfrak{v}(S \cup T) \succcurlyeq \mathfrak{v}(S) \oplus \mathfrak{v}(T).$$

Given a finite set $N = \{1, 2, \cdots, n\}$ of players, let $|S|$ denote the cardinality of a subset $S$ of $N$. Therefore we have $|N| = n$.

**Example 1.** *We consider a factory with n workers. Assume that each worker is doing the same task. Owing to the unexpected uncertainty and ability, the worker $i$ can earn around $A_i$ dollars, where the statement "around $A_i$ dollars" can be modeled as a bounded closed interval in $\mathbb{R}$. Therefore, we set $\mathfrak{v}(\{i\}) = A_i$ for $i \in N$. Given a coalition $S \subseteq N$, the cooperation for the workers in $S$ shows that there is an extra benefit $|S|B$ can be earned, where the benefit depends on the cardinality of the coalition $S$ and the benefit is "around B dollars" for each member.*

*In this case, the interval-valued payoff $\mathfrak{v}(S)$ can be defined as follows*

$$\mathfrak{v}(S) = \left( \bigoplus_{i \in S} A_i \right) \oplus (|S|B).$$

*It is also reasonable to define $v(\varnothing) = [0, 0]$.*

## 4. Cooperative Games with Interval-Valued Payoffs

We consider a finite set $N = \{1, 2, \cdots, n\}$ of players. The subsets of $N$ are called the *coalitions*. We denote by $\mathcal{P}(N)$ the collection of all coalitions. Let $\mathfrak{v} : \mathcal{P}(N) \to \mathcal{I}$ be an interval-valued function defined on $\mathcal{P}(N)$. Then $(N, \mathfrak{v})$ is called a *cooperative game with interval-valued payoff* if and only if $\mathfrak{v}(\emptyset) = [0, 0]$. Given a coalition $S$, owing to the unexpected situation, the payoff $\mathfrak{v}(S)$ regarding this coalition $S$ is assumed to be uncertain. In other words, the map $\mathfrak{v}$ assigns to each coalition a bounded closed interval telling what such a coalition can achieve in cooperation.

Let $(N, \mathfrak{v})$ be a cooperative game with interval-valued payoff. For any $S, T \subseteq N$ with $S \cap T = \emptyset$, we define three types of superadditivity as follows:

- We say that $(N, \mathfrak{v})$ is $\succeq$-*superadditive* if and only if

$$\mathfrak{v}(S \cup T) \succeq \mathfrak{v}(S) \oplus \mathfrak{v}(T);$$

- We say that $(N, \mathfrak{v})$ is $\succsim$-*superadditive* if and only if

$$\mathfrak{v}(S \cup T) \succsim \mathfrak{v}(S) \oplus \mathfrak{v}(T);$$

- We say that $(N, \mathfrak{v})$ is $\succcurlyeq$-*superadditive* if and only if

$$\mathfrak{v}(S \cup T) \succcurlyeq \mathfrak{v}(S) \oplus \mathfrak{v}(T).$$

Given a finite set $N = \{1, 2, \cdots, n\}$ of players, let $|S|$ denote the cardinality of a subset $S$ of $N$. Therefore we have $|N| = n$.

**Example 2.** *We consider a factory with n workers. Assume that each worker is doing the same task. Owing to the unexpected uncertainty and ability, the worker i can earn around $A_i$ dollars, where the statement "around $A_i$ dollars" can be modeled as a bounded closed interval in $\mathbb{R}$. Therefore, we set $\mathfrak{v}(\{i\}) = A_i$ for $i \in N$. Given a coalition $S \subseteq N$, the cooperation for the workers in S shows that there is an extra benefit $|S|B$ can be earned, where the benefit depends on the cardinality of the coalition S and the benefit is "around B dollars" for each member. In this case, the interval-valued payoff $\mathfrak{v}(S)$ can be defined as follows*

$$\mathfrak{v}(S) = \left( \bigoplus_{i \in S} A_i \right) \oplus (|S|B).$$

*It is also reasonable to define $v(\emptyset) = [0, 0]$.*

## 5. Cores of Cooperative Game with Interval-Valued Payoffs

Given a finite set $N = \{1, 2, \cdots, n\}$ of players, we write $\mathbf{A} \in \mathcal{I}^{|N|}$ to denote the $|N|$-dimensional vector in the product set $\mathcal{I}^{|N|}$ given by

$$\mathbf{A} = (A_1, A_2, \cdots, A_n) \in \mathcal{I} \times \mathcal{I} \times \cdots \times \mathcal{I}.$$

Let $(N, \mathfrak{v})$ be a cooperative game with interval-valued payoff. The set of *pre-imputation* of $(N, \mathfrak{v})$ is denoted and defined by

$$\mathfrak{I}^{\circ}(\mathfrak{v}) = \left\{ \mathbf{A} \in \mathcal{I}^{|N|} : \bigoplus_{i \in N} A_i =_{\Omega} \mathfrak{v}(N) \right\}$$

$$= \left\{ \mathbf{A} \in \mathcal{I}^{|N|} : \bigoplus_{i \in N} A_i \oplus \omega_1 = \mathfrak{v}(N) \oplus \omega_2 \text{ for some } \omega_1, \omega_2 \in \Omega \right\}.$$

Suppose that the individual rationality is considered. The different sets of *imputation* of $(N, \mathfrak{v})$ are defined by

$$\mathfrak{I}(\mathfrak{v}, \succeq) = \{\mathbf{A} \in \mathfrak{I}^\circ(\mathfrak{v}) : A_i \succeq \mathfrak{v}(\{i\}) \text{ for each } i \in N\}$$

and

$$\mathfrak{I}(\mathfrak{v}, \succeq_\Omega) = \{\mathbf{A} \in \mathfrak{I}^\circ(\mathfrak{v}) : A_i \succeq_\Omega \mathfrak{v}(\{i\}) \text{ for each } i \in N\}$$
$$= \{\mathbf{A} \in \mathfrak{I}^\circ(\mathfrak{v}) : \text{ for each } i \in N, A_i \oplus \omega_1 \succeq \mathfrak{v}(\{i\}) \oplus \omega_1 \text{ for some } \omega_1, \omega_2 \in \Omega\}.$$

We can similarly define $\mathfrak{I}(\mathfrak{v}, \succcurlyeq)$, $\mathfrak{I}(\mathfrak{v}, \succcurlyeq_\Omega)$, $\mathfrak{I}(\mathfrak{v}, \succsim)$ and $\mathfrak{I}(\mathfrak{v}, \succsim_\Omega)$ based on the different binary relations.

- It is clear that $\mathfrak{I}(\mathfrak{v}, \succeq) \subseteq \mathfrak{I}(\mathfrak{v}, \succeq_\Omega)$, $\mathfrak{I}(\mathfrak{v}, \succcurlyeq) \subseteq \mathfrak{I}(\mathfrak{v}, \succcurlyeq_\Omega)$ and $\mathfrak{I}(\mathfrak{v}, \succsim) \subseteq \mathfrak{I}(\mathfrak{v}, \succsim_\Omega)$.
- Using Remark 3, we see that $\mathfrak{I}(\mathfrak{v}, \succsim_\Omega) \subseteq \mathfrak{I}(\mathfrak{v}, \succeq_\Omega) \subseteq \mathfrak{I}(\mathfrak{v}, \succcurlyeq_\Omega)$.

**Example 3.** *Continued from Example 2, we have*

$$\mathfrak{v}(N) = \left(\bigoplus_{i \in N} A_i\right) \oplus |N|B.$$

*Therefore the pre-imputation is given by*

$$\mathfrak{I}^\circ(\mathfrak{v}) = \left\{\mathbf{A} \in \mathcal{I}^{|N|} : \bigoplus_{i \in N} A_i \oplus \omega_1 = \left(\bigoplus_{i \in N} A_i\right) \oplus |N|B \oplus \omega_2 \text{ for some } \omega_1, \omega_2 \in \Omega\right\}.$$

*The imputation of $(N, \mathfrak{v})$ are also given by*

$$\mathfrak{I}(\mathfrak{v}, \succcurlyeq) = \{\mathbf{A} \in \mathfrak{I}^\circ(\mathfrak{v}) : A_i \succcurlyeq A_i \text{ for each } i \in N\}$$

*and*

$$\mathfrak{I}(\mathfrak{v}, \succsim_\Omega) = \{\mathbf{A} \in \mathfrak{I}^\circ(\mathfrak{v}) : \text{ for each } i \in N, A_i \oplus \omega_1 \succsim A_i \oplus \omega_1 \text{ for some } \omega_1, \omega_2 \in \Omega\}.$$

Let $(N, \mathfrak{v})$ be a cooperative game with interval-valued payoff. In order to define the *core* of $(N, \mathfrak{v})$. We first define the concept of *anti-core*.

The different types of *anti-core* of $(N, \mathfrak{v})$ are denoted and defined as follows. Under the set of imputation $\mathfrak{I}(\mathfrak{v}, \succeq)$, we define

$$\widehat{\mathfrak{C}}(\mathfrak{v}, \succeq; \prec) = \left\{\mathbf{A} \in \mathfrak{I}(\mathfrak{v}, \succeq) : \text{there exists } S \subseteq N \text{ such that } \bigoplus_{i \in S} A_i \prec \mathfrak{v}(S)\right\}$$

and

$$\widehat{\mathfrak{C}}(\mathfrak{v}, \succeq; \prec_\Omega) = \left\{\mathbf{A} \in \mathfrak{I}(\mathfrak{v}, \succeq) : \text{there exists } S \subseteq N \text{ such that } \bigoplus_{i \in S} A_i \prec_\Omega \mathfrak{v}(S)\right\}.$$

Under the set of imputation $\mathfrak{I}(\mathfrak{v}, \succeq_\Omega)$, we define

$$\widehat{\mathfrak{C}}(\mathfrak{v}, \succeq_\Omega; \prec) = \left\{\mathbf{A} \in \mathfrak{I}(\mathfrak{v}, \succeq_\Omega) : \text{there exists } S \subseteq N \text{ such that } \bigoplus_{i \in S} A_i \prec \mathfrak{v}(S)\right\}$$

and

$$\widehat{\mathfrak{C}}(\mathfrak{v}, \succeq_\Omega; \prec_\Omega) = \left\{\mathbf{A} \in \mathfrak{I}(\mathfrak{v}, \succeq_\Omega) : \text{there exists } S \subseteq N \text{ such that } \bigoplus_{i \in S} A_i \prec_\Omega \mathfrak{v}(S)\right\}.$$

Since $A \lessdot B$ implies $A \lessdot_\Omega B$, we have the inclusion

$$\widehat{\mathfrak{C}}(\mathfrak{v}, \succeq; \lessdot) \subseteq \widehat{\mathfrak{C}}(\mathfrak{v}, \succeq; \lessdot_\Omega) \text{ and } \widehat{\mathfrak{C}}(\mathfrak{v}, \succeq_\Omega; \lessdot) \subseteq \widehat{\mathfrak{C}}(\mathfrak{v}, \succeq_\Omega; \lessdot_\Omega).$$

Based on the different binary relations, we can similarly define the other types of anti-core. For example, it is clear to see that

$$\widehat{\mathfrak{C}}(\mathfrak{v}, \succeq; \prec) = \left\{ \mathbf{A} \in \mathfrak{I}(\mathfrak{v}, \succeq) : \text{there exists } S \subseteq N \text{ such that } \bigoplus_{i \in S} A_i \prec \mathfrak{v}(S) \right\}$$

$$\widehat{\mathfrak{C}}(\mathfrak{v}, \succcurlyeq_\Omega; \lessdot_\Omega) = \left\{ \mathbf{A} \in \mathfrak{I}(\mathfrak{v}, \succcurlyeq_\Omega) : \text{there exists } S \subseteq N \text{ such that } \bigoplus_{i \in S} A_i \lessdot_\Omega \mathfrak{v}(S) \right\}$$

$$\widehat{\mathfrak{C}}(\mathfrak{v}, \succsim; \prec_\Omega) = \left\{ \mathbf{A} \in \mathfrak{I}(\mathfrak{v}, \succsim) : \text{there exists } S \subseteq N \text{ such that } \bigoplus_{i \in S} A_i \prec_\Omega \mathfrak{v}(S) \right\}.$$

Now the *core* of $(N, \mathfrak{v})$ is defined to be the complement set of anti-core with respect to the corresponding set of imputation. More precisely, the *core* $\mathfrak{C}(\mathfrak{v}, \succeq; \lessdot)$ of $(N, \mathfrak{v})$ is denoted and defined by

$$\mathfrak{C}(\mathfrak{v}, \succeq; \lessdot) = \mathfrak{I}(\mathfrak{v}, \succeq) \setminus \widehat{\mathfrak{C}}(\mathfrak{v}, \succeq; \lessdot),$$

and the core $\mathfrak{C}(\mathfrak{v}, \succeq; \lessdot_\Omega)$ of $(N, \mathfrak{v})$ is denoted and defined by

$$\mathfrak{C}(\mathfrak{v}, \succeq; \lessdot_\Omega) = \mathfrak{I}(\mathfrak{v}, \succeq) \setminus \widehat{\mathfrak{C}}(\mathfrak{v}, \succeq; \lessdot_\Omega).$$

It is also clear to see that

$$\mathfrak{C}(\mathfrak{v}, \succsim; \lessdot_\Omega) = \mathfrak{I}(\mathfrak{v}, \succsim) \setminus \widehat{\mathfrak{C}}(\mathfrak{v}, \succsim; \lessdot_\Omega) \text{ and } \mathfrak{C}(\mathfrak{v}, \succcurlyeq_\Omega; \lessdot) = \mathfrak{I}(\mathfrak{v}, \succcurlyeq_\Omega) \setminus \widehat{\mathfrak{C}}(\mathfrak{v}, \succcurlyeq_\Omega; \lessdot).$$

Equivalently, we also see that

$$\mathfrak{C}(\mathfrak{v}, \succeq; \lessdot) = \left\{ \mathbf{A} \in \mathfrak{I}(\mathfrak{v}, \succeq) : \bigoplus_{i \in S} A_i \not\lessdot \mathfrak{v}(S) \text{ for all } S \subseteq N \right\}$$

and

$$\mathfrak{C}(\mathfrak{v}, \succeq_\Omega; \lessdot) = \left\{ \mathbf{A} \in \mathfrak{I}(\mathfrak{v}, \succeq_\Omega) : \bigoplus_{i \in S} A_i \not\lessdot \mathfrak{v}(S) \text{ for all } S \subseteq N \right\}.$$

Given any two bounded closed intervals $A$ and $B$. If there exists another bounded closed interval $C$ satisfying $A = B \oplus C$, then we say that the *Hukuhara difference* between $A$ and $B$ exists, and we write $C = A \ominus_H B$. It is clear to see that $c^L = a^L - b^L$ and $c^U = a^U - b^U$.

Based on the Hukuhara difference, we also want to define the anti-cores and core of a cooperative game with interval-valued payoff. Under the set of imputation $\mathfrak{I}(\mathfrak{v}, \succeq)$, the *H-anti-cores* of $(N, \mathfrak{v})$ are denoted and defined by

$$\widehat{\mathfrak{C}}_H(\mathfrak{v}, \succeq; \prec) = \left\{ \mathbf{A} \in \mathfrak{I}(\mathfrak{v}, \succeq) : \text{there exists } S \subseteq N \text{ such that } \bigoplus_{i \in S} A_i \prec \mathfrak{v}(S) \right.$$

$$\left. \text{and that the Hukuhara difference } \mathfrak{v}(S) \ominus_H \bigoplus_{i \in S} A_i \text{ exists} \right\}$$

and

$$\widehat{\mathfrak{C}}_H(\mathfrak{v}, \succeq; \prec_\Omega) = \{\mathbf{A} \in \mathfrak{I}(\mathfrak{v}, \succeq) : \text{there exist } S \subseteq N \text{ and } \omega_1, \omega_2 \in \Omega \text{ such that}$$

$$\bigoplus_{i \in S} A_i \oplus \omega_1 \prec \mathfrak{v}(S) \oplus \omega_2 \text{ and that the Hukuhara difference}$$

$$(\mathfrak{v}(S) \oplus \omega_2) \ominus_H \left(\bigoplus_{i \in S} A_i \oplus \omega_1\right) \text{ exists}\bigg\}.$$

The *H-cores* of $(N, \mathfrak{v})$ are denoted and defined by

$$\mathfrak{C}_H(\mathfrak{v}, \succeq; \prec) = \mathfrak{I}(\mathfrak{v}, \succeq) \setminus \widehat{\mathfrak{C}}_H(\mathfrak{v}, \succeq; \prec) \text{ and } \mathfrak{C}_H(\mathfrak{v}, \succeq; \prec_\Omega) = \mathfrak{I}(\mathfrak{v}, \succeq) \setminus \widehat{\mathfrak{C}}_H(\mathfrak{v}, \succeq; \prec_\Omega).$$

More precisely, we have

$$\mathfrak{C}_H(\mathfrak{v}, \succeq; \prec) = \left\{\mathbf{A} \in \mathfrak{I}(\mathfrak{v}, \succeq) : \text{for each } S \subseteq N, \text{ either } \bigoplus_{i \in S} A_i \not\prec \mathfrak{v}(S)\right.$$

$$\left.\text{or the Hukuhara difference } \mathfrak{v}(S) \ominus_H \bigoplus_{i \in S} A_i \text{ does not exist}\right\}$$

and

$$\mathfrak{C}_H(\mathfrak{v}, \succeq; \prec_\Omega) = \{\mathbf{A} \in \mathfrak{I}(\mathfrak{v}, \succeq) : \text{for each } S \subseteq N \text{ and for any } \omega_1, \omega_2 \in \Omega,$$

$$\text{either } \bigoplus_{i \in S} A_i \oplus \omega_1 \not\prec \mathfrak{v}(S) \oplus \omega_2 \text{ or the Hukuhara difference}$$

$$(\mathfrak{v}(S) \oplus \omega_2) \ominus_H \left(\bigoplus_{i \in S} A_i \oplus \omega_1\right) \text{ does not exist}\bigg\}.$$

Based on the different binary relations, we can similarly define the other types of H-core of $(N, \mathfrak{v})$. Using Remarks 3 and 4, we can obtain many inclusions regarding the different types of cores and H-cores. We omit the details.

Without considering the individual rationality, we can similarly define the *pre-core* $\mathfrak{C}^\circ$ and *pre-H-core* $\mathfrak{C}_H^\circ$ based on the pre-imputation $\mathfrak{I}^\circ$.

**Example 4.** *Continued from Example 3, if* $\mathbf{A} \in \mathfrak{C}_H(\mathfrak{v}, \succeq; \prec)$, *then, for each* $S \subseteq N$, *either*

$$\bigoplus_{i \in S} A_i \not\prec \left[\left(\bigoplus_{i \in S} A_i\right) \oplus (|S|B)\right]$$

*or the Hukuhara difference*

$$\left[\left(\bigoplus_{i \in S} A_i\right) \oplus (|S|B)\right] \ominus_H \bigoplus_{i \in S} A_i$$

*does not exists.*

## 6. Dominance Cores of Cooperative Game with Interval-Valued Payoffs

Let $(N, \mathfrak{v})$ be a cooperative game with interval-valued payoff. We are going to define many types of dominance core and H-dominance core.

- For $\mathbf{A}, \mathbf{B} \in \mathfrak{I}(\mathfrak{v}, \succeq)$ and $S \subseteq N$, we say that $\mathbf{B}$ $(\succeq; \prec, \preceq_\Omega)$-*dominates* $\mathbf{A}$ *via coalition S* if and only if $A_i \prec B_i$ for each $i \in S$ and

$$\bigoplus_{i \in S} B_i \preceq_\Omega \mathfrak{v}(S).$$

We simply say that **B** $(\succeq; \prec, \preceq_\Omega)$-dominates **A** if and only if there is a coalition $S$ such that **B** $(\succeq; \prec, \preceq_\Omega)$-dominates **A** via $S$. The $(\succeq; \prec, \preceq_\Omega)$-*dominance core* of $(N, \mathfrak{v})$, denoted by $\mathfrak{DC}(\mathfrak{v}, \succeq; \prec, \preceq_\Omega)$, is the set of imputation $\mathfrak{I}(\mathfrak{v}, \succeq)$ that are $(\succeq; \prec, \preceq_\Omega)$-nondominated.

- For **A**, **B** $\in \mathfrak{I}(\mathfrak{v}, \succeq)$ and $S \subseteq N$, we say that **B** $(\succeq; \prec, \preceq)$-*H-dominates* **A** *via coalition S* if and only if $A_i \prec B_i$ for each $i \in S$ and

$$\bigoplus_{i \in S} B_i \preceq \mathfrak{v}(S) \text{ and the Hukuhara difference } \mathfrak{v}(S) \ominus_H \left( \bigoplus_{i \in S} A_i \right) \text{ exists.}$$

We simply say that **B** $(\succeq; \prec, \preceq)$-H-dominates **A** if and only if there is a coalition $S$ such that **B** $(\succeq; \prec, \preceq)$-H-dominates **A** via $S$. The $(\succeq; \prec, \preceq)$-*H-dominance core* of $(N, \mathfrak{v})$, denoted by $\mathfrak{DC}_H(\mathfrak{v}, \succeq; \prec, \preceq)$, is the set of imputation $\mathfrak{I}(\mathfrak{v}, \succeq)$ that are $(\succeq; \prec, \preceq)$-H-nondominated.

- For **A**, **B** $\in \mathfrak{I}(\mathfrak{v}, \succeq)$ and $S \subseteq N$, we say that **B** $(\succeq; \prec, \preceq_\Omega)$-*H-dominates* **A** *via coalition S* if and only if $A_i \prec B_i$ for each $i \in S$, and there exist $\omega_1, \omega_2 \in \Omega$ such that

$$\bigoplus_{i \in S} B_i \oplus \omega_1 \preceq \mathfrak{v}(S) \oplus \omega_2$$

and the Hukuhara difference

$$(\mathfrak{v}(S) \oplus \omega_2) \ominus_H \left( \bigoplus_{i \in S} A_i \oplus \omega_1 \right) \text{ exists.}$$

We simply say that **B** $(\succeq; \prec, \preceq_\Omega)$-H-dominates **A** if and only if there is a coalition $S$ such that **B** $(\succeq; \prec, \preceq_\Omega)$-H-dominates **A** via $S$. The $(\succeq; \prec, \preceq_\Omega)$-*H-dominance core* of $(N, \mathfrak{v})$, denoted by $\mathfrak{DC}_H(\mathfrak{v}, \succeq; \prec, \preceq_\Omega)$, is the set of imputation $\mathfrak{I}(\mathfrak{v}, \succeq)$ that are $(\succeq; \prec, \preceq_\Omega)$-H-nondominated.

Based on the different binary relations, we can similarly define the other types of dominance cores and H-dominance cores. For example, it is clear to see that $\mathfrak{DC}(\mathfrak{v}, \succsim_\Omega; \lll, \preceq_\Omega)$ is the set of imputation $\mathfrak{I}(\mathfrak{v}, \succsim_\Omega)$ that are $(\succsim_\Omega; \lll, \preceq_\Omega)$-nondominated, and $\mathfrak{DC}_H(\mathfrak{v}, \succcurlyeq; \lll, \preccurlyeq_\Omega)$ is the set of imputation $\mathfrak{I}(\mathfrak{v}, \succcurlyeq)$ that are $(\succcurlyeq; \lll, \preccurlyeq_\Omega)$-H-nondominated. Using Remarks 3 and 4, we can obtain many inclusions regarding the different types of dominance cores and H-dominance cores. We omit the details.

Without considering the individual rationality, we can similarly define the *dominance pre-core* $\mathfrak{DC}^\circ$ and *H-dominance pre-core* $\mathfrak{DC}_H^\circ$ based on the pre-imputation $\mathfrak{I}^\circ$.

**Example 5.** *Continued from Example 3, for* **A**, **B** $\in \mathfrak{I}(\mathfrak{v}, \succeq)$ *and* $S \subseteq N$, *we see that* **B** $(\succeq; \prec, \preceq)$-*H-dominates* **A** *via coalition S if and only if* $A_i \prec B_i$ *for each* $i \in S$ *and*

$$\bigoplus_{i \in S} B_i \preceq \left[ \left( \bigoplus_{i \in S} A_i \right) \oplus (|S|B) \right]$$

*and the Hukuhara difference*

$$\left[ \left( \bigoplus_{i \in S} A_i \right) \oplus (|S|B) \right] \ominus_H \left( \bigoplus_{i \in S} A_i \right) \text{ exists.}$$

## 7. The Relations between Cores and Dominance Cores

We are going to investigate the inclusions and equalities between cores (resp. H-cores) and dominance cores (resp. H-dominance cores).

**Proposition 3.** *Given a cooperative game with interval-valued payoff* $(N, \mathfrak{v})$, *under the set of imputation* $\mathfrak{I}(\mathfrak{v}, \succeq)$, *we have the following inclusions:*

- $\mathfrak{C}(\mathfrak{v}, \succeq; \lessdot_\Omega) \subseteq \mathfrak{DC}(\mathfrak{v}, \succeq; \sqsubset, \precsim_\Omega)$ *and* $\mathfrak{C}(\mathfrak{v}, \succeq; \lessdot) \subseteq \mathfrak{DC}(\mathfrak{v}, \succeq; \sqsubset, \precsim)$;
- $\mathfrak{C}(\mathfrak{v}, \succeq; \sqsubset_\Omega) \subseteq \mathfrak{DC}(\mathfrak{v}, \succeq; \sqsubset, \preccurlyeq_\Omega)$ *and* $\mathfrak{C}(\mathfrak{v}, \succeq; \sqsubset) \subseteq \mathfrak{DC}(\mathfrak{v}, \succeq; \sqsubset, \preccurlyeq)$;
- $\mathfrak{C}(\mathfrak{v}, \succeq; \sqsubset_\Omega) \subseteq \mathfrak{DC}(\mathfrak{v}, \succeq; \sqsubset, \preceq_\Omega)$ *and* $\mathfrak{C}(\mathfrak{v}, \succeq; \sqsubset) \subseteq \mathfrak{DC}(\mathfrak{v}, \succeq; \sqsubset, \preceq)$;
- $\mathfrak{C}(\mathfrak{v}, \succeq; \prec_\Omega) \subseteq \mathfrak{DC}(\mathfrak{v}, \succeq; \prec, \preceq_\Omega)$ *and* $\mathfrak{C}(\mathfrak{v}, \succeq; \prec) \subseteq \mathfrak{DC}(\mathfrak{v}, \succeq; \prec, \preceq)$;
- $\mathfrak{C}(\mathfrak{v}, \succeq; \sqsubset_\Omega) \subseteq \mathfrak{DC}(\mathfrak{v}, \succeq; \prec, \preccurlyeq_\Omega)$ *and* $\mathfrak{C}(\mathfrak{v}, \succeq; \sqsubset) \subseteq \mathfrak{DC}(\mathfrak{v}, \succeq; \prec, \preccurlyeq)$;
- $\mathfrak{C}(\mathfrak{v}, \succeq; \lessdot_\Omega) \subseteq \mathfrak{DC}(\mathfrak{v}, \succeq; \prec, \precsim_\Omega)$ *and* $\mathfrak{C}(\mathfrak{v}, \succeq; \lessdot) \subseteq \mathfrak{DC}(\mathfrak{v}, \succeq; \prec, \precsim)$;
- $\mathfrak{C}(\mathfrak{v}, \succeq; \nprec_\Omega) \subseteq \mathfrak{DC}(\mathfrak{v}, \succeq; \nprec, \preceq_\Omega)$ *and* $\mathfrak{C}(\mathfrak{v}, \succeq; \nprec) \subseteq \mathfrak{DC}(\mathfrak{v}, \succeq; \nprec, \preceq)$;
- $\mathfrak{C}(\mathfrak{v}, \succeq; \lessdot_\Omega) \subseteq \mathfrak{DC}(\mathfrak{v}, \succeq; \nprec, \precsim_\Omega)$ *and* $\mathfrak{C}(\mathfrak{v}, \succeq; \lessdot) \subseteq \mathfrak{DC}(\mathfrak{v}, \succeq; \nprec, \precsim)$;
- $\mathfrak{C}(\mathfrak{v}, \succeq; \sqsubset_\Omega) \subseteq \mathfrak{DC}(\mathfrak{v}, \succeq; \nprec, \preccurlyeq_\Omega)$ *and* $\mathfrak{C}(\mathfrak{v}, \succeq; \sqsubset) \subseteq \mathfrak{DC}(\mathfrak{v}, \succeq; \nprec, \preccurlyeq)$;
- $\mathfrak{C}(\mathfrak{v}, \succeq; \prec_\Omega) \subseteq \mathfrak{DC}(\mathfrak{v}, \succeq; \lessdot, \preceq_\Omega)$ *and* $\mathfrak{C}(\mathfrak{v}, \succeq; \prec) \subseteq \mathfrak{DC}(\mathfrak{v}, \succeq; \lessdot, \preceq)$;
- $\mathfrak{C}(\mathfrak{v}, \succeq; \sqsubset_\Omega) \subseteq \mathfrak{DC}(\mathfrak{v}, \succeq; \lessdot, \preccurlyeq_\Omega)$ *and* $\mathfrak{C}(\mathfrak{v}, \succeq; \sqsubset) \subseteq \mathfrak{DC}(\mathfrak{v}, \succeq; \lessdot, \preccurlyeq)$;
- $\mathfrak{C}(\mathfrak{v}, \succeq; \lessdot_\Omega) \subseteq \mathfrak{DC}(\mathfrak{v}, \succeq; \lessdot, \precsim_\Omega)$ *and* $\mathfrak{C}(\mathfrak{v}, \succeq; \lessdot) \subseteq \mathfrak{DC}(\mathfrak{v}, \succeq; \lessdot, \precsim)$;

**Proof.** We first prove the inclusion $\mathfrak{C}(\mathfrak{v}, \succeq, \lessdot_\Omega) \subseteq \mathfrak{DC}(\mathfrak{v}, \succeq; \nprec, \precsim_\Omega)$. The inclusion is obvious when $\mathfrak{I}(\mathfrak{v}, \succeq) = \varnothing$.

Therefore we assume $\mathfrak{I}(\mathfrak{v}, \succeq) \neq \varnothing$. For $\mathbf{A} \in \mathfrak{I}(\mathfrak{v}, \succeq) \setminus \mathfrak{DC}(\mathfrak{v}, \succeq; \nprec, \precsim_\Omega)$, there exists $\mathbf{B} \in \mathfrak{I}(\mathfrak{v}, \succeq)$ such that $\mathbf{B}$ $(\succeq; \nprec, \precsim_\Omega)$-dominates $\mathbf{A}$.

By definition, there exists $S \subseteq N$ satisfying $A_i \nprec B_i$ for each $i \in S$ and

$$\bigoplus_{i \in S} B_i \precsim_\Omega \mathfrak{v}(S).$$

Therefore we obtain

$$\bigoplus_{i \in S} A_i \nprec \bigoplus_{i \in S} B_i \precsim_\Omega \mathfrak{v}(S),$$

which says that $\bigoplus_{i \in S} A_i \lessdot_\Omega \mathfrak{v}(S)$ by Remark 5, i.e.,

$$\mathbf{A} \in \widehat{\mathfrak{C}}(\mathfrak{v}; \succeq, \lessdot_\Omega) = \mathfrak{I}(\mathfrak{v}, \succeq) \setminus \mathfrak{C}(\mathfrak{v}; \succeq, \lessdot_\Omega).$$

We can similarly obtain the inclusion $\mathfrak{C}(\mathfrak{v}, \succeq, \lessdot) \subseteq \mathfrak{DC}(\mathfrak{v}, \succeq; \nprec, \precsim)$. The other types of inclusions can be similarly obtained by Proposition 2 and Remark 5. This completes the proof. $\square$

**Proposition 4.** *Given a cooperative game with interval-valued payoff* $(N, \mathfrak{v})$, *under the set of imputation* $\mathfrak{I}(\mathfrak{v}, \succeq)$, *we have the following inclusions:*

- $\mathfrak{C}_H(\mathfrak{v}, \succeq; \lessdot_\Omega) \subseteq \mathfrak{DC}_H(\mathfrak{v}, \succeq; \sqsubset, \precsim_\Omega)$ *and* $\mathfrak{C}_H(\mathfrak{v}, \succeq; \lessdot) \subseteq \mathfrak{DC}_H(\mathfrak{v}, \succeq; \sqsubset, \precsim)$;
- $\mathfrak{C}_H(\mathfrak{v}, \succeq; \sqsubset_\Omega) \subseteq \mathfrak{DC}_H(\mathfrak{v}, \succeq; \sqsubset, \preccurlyeq_\Omega)$ *and* $\mathfrak{C}_H(\mathfrak{v}, \succeq; \sqsubset) \subseteq \mathfrak{DC}_H(\mathfrak{v}, \succeq; \sqsubset, \preccurlyeq)$;
- $\mathfrak{C}_H(\mathfrak{v}, \succeq; \sqsubset_\Omega) \subseteq \mathfrak{DC}_H(\mathfrak{v}, \succeq; \sqsubset, \preceq_\Omega)$ *and* $\mathfrak{C}_H(\mathfrak{v}, \succeq; \sqsubset) \subseteq \mathfrak{DC}_H(\mathfrak{v}, \succeq; \sqsubset, \preceq)$;
- $\mathfrak{C}_H(\mathfrak{v}, \succeq; \prec_\Omega) \subseteq \mathfrak{DC}_H(\mathfrak{v}, \succeq; \prec, \preceq_\Omega)$ *and* $\mathfrak{C}_H(\mathfrak{v}, \succeq; \prec) \subseteq \mathfrak{DC}_H(\mathfrak{v}, \succeq; \prec, \preceq)$;
- $\mathfrak{C}_H(\mathfrak{v}, \succeq; \sqsubset_\Omega) \subseteq \mathfrak{DC}_H(\mathfrak{v}, \succeq; \prec, \preccurlyeq_\Omega)$ *and* $\mathfrak{C}_H(\mathfrak{v}, \succeq; \sqsubset) \subseteq \mathfrak{DC}_H(\mathfrak{v}, \succeq; \prec, \preccurlyeq)$;
- $\mathfrak{C}_H(\mathfrak{v}, \succeq; \lessdot_\Omega) \subseteq \mathfrak{DC}_H(\mathfrak{v}, \succeq; \prec, \precsim_\Omega)$ *and* $\mathfrak{C}_H(\mathfrak{v}, \succeq; \lessdot) \subseteq \mathfrak{DC}_H(\mathfrak{v}, \succeq; \prec, \precsim)$;
- $\mathfrak{C}_H(\mathfrak{v}, \succeq; \nprec_\Omega) \subseteq \mathfrak{DC}_H(\mathfrak{v}, \succeq; \nprec, \preceq_\Omega)$ *and* $\mathfrak{C}_H(\mathfrak{v}, \succeq; \nprec) \subseteq \mathfrak{DC}_H(\mathfrak{v}, \succeq; \nprec, \preceq)$;
- $\mathfrak{C}_H(\mathfrak{v}, \succeq; \lessdot_\Omega) \subseteq \mathfrak{DC}_H(\mathfrak{v}, \succeq; \nprec, \precsim_\Omega)$ *and* $\mathfrak{C}_H(\mathfrak{v}, \succeq; \lessdot) \subseteq \mathfrak{DC}_H(\mathfrak{v}, \succeq; \nprec, \precsim)$;
- $\mathfrak{C}_H(\mathfrak{v}, \succeq; \sqsubset_\Omega) \subseteq \mathfrak{DC}_H(\mathfrak{v}, \succeq; \nprec, \preccurlyeq_\Omega)$ *and* $\mathfrak{C}_H(\mathfrak{v}, \succeq; \sqsubset) \subseteq \mathfrak{DC}_H(\mathfrak{v}, \succeq; \nprec, \preccurlyeq)$;
- $\mathfrak{C}_H(\mathfrak{v}, \succeq; \prec_\Omega) \subseteq \mathfrak{DC}_H(\mathfrak{v}, \succeq; \lessdot, \preceq_\Omega)$ *and* $\mathfrak{C}_H(\mathfrak{v}, \succeq; \prec) \subseteq \mathfrak{DC}_H(\mathfrak{v}, \succeq; \lessdot, \preceq)$;
- $\mathfrak{C}_H(\mathfrak{v}, \succeq; \sqsubset_\Omega) \subseteq \mathfrak{DC}_H(\mathfrak{v}, \succeq; \lessdot, \preccurlyeq_\Omega)$ *and* $\mathfrak{C}_H(\mathfrak{v}, \succeq; \sqsubset) \subseteq \mathfrak{DC}_H(\mathfrak{v}, \succeq; \lessdot, \preccurlyeq)$;
- $\mathfrak{C}_H(\mathfrak{v}, \succeq; \lessdot_\Omega) \subseteq \mathfrak{DC}_H(\mathfrak{v}, \succeq; \lessdot, \precsim_\Omega)$ *and* $\mathfrak{C}_H(\mathfrak{v}, \succeq; \lessdot) \subseteq \mathfrak{DC}_H(\mathfrak{v}, \succeq; \lessdot, \precsim)$;

**Proof.** We first prove the inclusion $\mathfrak{C}_H(\mathfrak{v}, \succeq; \prec) \subseteq \mathfrak{DC}_H(\mathfrak{v}, \succeq; \prec, \preceq)$. The inclusion is obvious when $\mathfrak{I}(\mathfrak{v}, \succeq) = \varnothing$. Therefore we assume $\mathfrak{I}(\mathfrak{v}, \succeq) \neq \varnothing$. For $\mathbf{A} \in \mathfrak{I}(\mathfrak{v}, \succeq) \setminus \mathfrak{DC}_H(\mathfrak{v}, \succeq; \prec, \preceq)$, there exists

$\mathbf{B} \in \mathfrak{I}(\mathfrak{v}, \succeq)$ such that $\mathbf{B}$ $(\succeq; \prec, \preceq)$-H-dominates $\mathbf{A}$. By definition, there exists $S \subseteq N$ satisfying $A_i \prec B_i$ for each $i \in S$,

$$\bigoplus_{i \in S} B_i \preceq \mathfrak{v}(S) \text{ and the Hukuhara difference } \mathfrak{v}(S) \ominus_H \left( \bigoplus_{i \in S} A_i \right) \text{ exists.}$$

Therefore we obtain

$$\bigoplus_{i \in S} A_i \prec \bigoplus_{i \in S} B_i \preceq \mathfrak{v}(S),$$

which says that $\bigoplus_{i \in S} A_i \prec \mathfrak{v}(S)$ by Proposition 2, i.e.,

$$\mathbf{A} \in \widehat{\mathfrak{C}}_H(\mathfrak{v}, \succeq; \prec) = \mathfrak{I}(\mathfrak{v}, \succeq) \setminus \mathfrak{C}_H(\mathfrak{v}, \succeq; \prec).$$

The other types of inclusions can be similarly obtained by Proposition 2 and Remark 5. This completes the proof. □

The inclusions in Propositions 3 and 4 are under the set of imputation $\mathfrak{I}(\mathfrak{v}, \succeq)$. We can obtain the similar inclusions under the different sets of imputation. We omit the details.

**Theorem 1.** *Let* $(N, \mathfrak{v})$ *be a cooperative game with an interval-valued payoff. Suppose that*

$$\mathfrak{v}(N) \succsim \mathfrak{v}(S) \oplus \left( \bigoplus_{i \in N \setminus S} \mathfrak{v}(\{i\}) \right) \tag{1}$$

*for each* $S \subseteq N$ *with* $S \neq N$. *Then*

$$\mathfrak{DC}(\mathfrak{v}, \succeq; \lll, \preceq_\Omega) \subseteq \mathfrak{C}(\mathfrak{v}, \succeq; \lll_\Omega) \text{ and } \mathfrak{DC}(\mathfrak{v}, \succeq; \lll, \preceq_\Omega) \subseteq \mathfrak{C}(\mathfrak{v}, \succeq; \lll).$$

**Proof.** For proving the inclusion $\mathfrak{DC}(\mathfrak{v}, \succeq; \lll, \preceq_\Omega) \subseteq \mathfrak{C}(\mathfrak{v}, \succeq; \lll_\Omega)$, we want to show that

$$\mathbf{A} \in \mathfrak{I}(\mathfrak{v}, \succeq) \setminus \mathfrak{C}(\mathfrak{v}, \succeq; \lll_\Omega) = \widehat{\mathfrak{C}}(\mathfrak{v}, \succeq; \lll_\Omega) \text{ implies } \mathbf{A} \in \mathfrak{I}(\mathfrak{v}, \succeq) \setminus \mathfrak{DC}(\mathfrak{v}, \succeq; \lll, \preceq_\Omega).$$

For $\mathbf{A} \in \widehat{\mathfrak{C}}(\mathfrak{v}, \succeq; \lll_\Omega)$, by the definition of anti-core, there exists $S \subseteq N$ such that $\bigoplus_{i \in S} A_i \lll_\Omega \mathfrak{v}(S)$, i.e., $\bigoplus_{i \in S} A_i \oplus \omega_1 \lll \mathfrak{v}(S) \oplus \omega_2$ for some $\omega_1, \omega_2 \in \Omega$. Let

$$\epsilon = (\mathfrak{v}(S) \oplus \omega_2) \ominus \left( \bigoplus_{i \in S} A_i \oplus \omega_1 \right).$$

Then

$$\epsilon^L = (\mathfrak{v}(S) \oplus \omega_2)^L - \left( \bigoplus_{i \in S} A_i \oplus \omega_1 \right)^U \text{ and } \epsilon^U = (\mathfrak{v}(S) \oplus \omega_2)^U - \left( \bigoplus_{i \in S} A_i \oplus \omega_1 \right)^L.$$

Since $\bigoplus_{i \in S} A_i \oplus \omega_1 \lll \mathfrak{v}(S) \oplus \omega_2$, i.e., $(\mathfrak{v}(S) \oplus \omega_2)^L > (\bigoplus_{i \in S} A_i \oplus \omega_1)^U$, it follows that $\epsilon^U \geq \epsilon^L > 0$. Using part (ii) of Proposition 1, we obtain

$$\epsilon \oplus \left( \bigoplus_{i \in S} A_i \right) =_\Omega \mathfrak{v}(S), \tag{2}$$

We are going to find an imputation $\mathbf{B}$ such that $\mathbf{B}$ $(\succeq; \lll, \preceq_\Omega)$-dominates $\mathbf{A}$ via $S$.

Suppose that $S \neq N$, i.e., $N \setminus S \neq \varnothing$, we define

$$B_i = \begin{cases} A_i \oplus \frac{1}{|S|}\epsilon, & \text{if } i \in S \\ \mathfrak{v}(\{i\}) \oplus \left( \frac{1}{|N \setminus S|} \left[ \mathfrak{v}(N) \ominus \left( \bigoplus_{i \in N \setminus S} \mathfrak{v}(\{i\}) \right) \ominus \mathfrak{v}(S) \right] \right), & \text{if } i \in N \setminus S. \end{cases}$$

Then, according to (2), we have

$$\bigoplus_{i \in S} B_i = \left( \bigoplus_{i \in S} A_i \right) \oplus \epsilon =_\Omega \mathfrak{v}(S),$$

which says that

$$\bigoplus_{i \in S} B_i \oplus \omega_3 = \mathfrak{v}(S) \oplus \omega_4 \tag{3}$$

for some $\omega_3, \omega_4 \in \Omega$. We also have

$$\bigoplus_{i \in N \setminus S} B_i = \bigoplus_{i \in N \setminus S} \mathfrak{v}(\{i\}) \oplus \left[ \mathfrak{v}(N) \ominus \left( \bigoplus_{i \in N \setminus S} \mathfrak{v}(\{i\}) \right) \ominus \mathfrak{v}(S) \right]. \tag{4}$$

Let

$$\omega_5 = \left( \bigoplus_{i \in N \setminus S} \mathfrak{v}(\{i\}) \right) \ominus \left( \bigoplus_{i \in N \setminus S} \mathfrak{v}(\{i\}) \right) \in \Omega \text{ and } \omega_6 = \mathfrak{v}(S) \ominus \mathfrak{v}(S) \in \Omega. \tag{5}$$

Then, from (3)–(5), we obtain

$$\bigoplus_{i \in N} B_i \oplus \omega_3 = \omega_3 \oplus \bigoplus_{i \in S} B_i \oplus \bigoplus_{i \in N \setminus S} B_i = \omega_4 \oplus \mathfrak{v}(S) \oplus \omega_5 \oplus \mathfrak{v}(N) \ominus \mathfrak{v}(S)$$

$$= \omega_4 \oplus \omega_5 \oplus \omega_6 \oplus \mathfrak{v}(N),$$

which says that

$$\bigoplus_{i \in N} B_i =_\Omega \mathfrak{v}(N)$$

by the fact of $\omega_4 \oplus \omega_5 \oplus \omega_6 \in \Omega$. Since $\mathbf{A} \in \mathfrak{I}(\mathfrak{v}, \succeq)$ and $\epsilon^U \geq \epsilon^L > 0$, we have

$$B_i = A_i \oplus \left( \frac{1}{|S|}\epsilon \right) \succ\succ A_i \succeq \mathfrak{v}(\{i\}) \text{ for each } i \in S,$$

which also implies $B_i \succ\succ \mathfrak{v}(\{i\})$, i.e., $B_i \succeq \mathfrak{v}(\{i\})$ for $i \in S$. Let

$$C \equiv \mathfrak{v}(N) \ominus \left( \bigoplus_{i \in N \setminus S} \mathfrak{v}(\{i\}) \right) \ominus \mathfrak{v}(S) = \mathfrak{v}(N) \ominus \left[ \mathfrak{v}(S) \oplus \left( \bigoplus_{i \in N \setminus S} \mathfrak{v}(\{i\}) \right) \right].$$

Then

$$c^L = (\mathfrak{v}(N))^L - \left( \mathfrak{v}(S) \oplus \left( \bigoplus_{i \in N \setminus S} \mathfrak{v}(\{i\}) \right) \right)^U$$

and

$$c^U = (\mathfrak{v}(N))^U - \left( \mathfrak{v}(S) \oplus \left( \bigoplus_{i \in N \setminus S} \mathfrak{v}(\{i\}) \right) \right)^L.$$

From (1), we see that $c^U \geq c^L \geq 0$, which says that $B_i \succeq \mathfrak{v}(\{i\})$ for $i \in N \setminus S$. From (3), we also have

$$\bigoplus_{i \in S} B_i \preceq_\Omega \mathfrak{v}(S).$$

This shows that $\mathbf{B} \in \mathfrak{I}(\mathfrak{v}, \succeq)$ and $\mathbf{B}$ $(\succeq; \lll, \preceq_\Omega)$-dominates $\mathbf{A}$ via $S$.

Suppose that $S = N$, for each $i \in N$, we define

$$B_i = A_i \oplus \frac{1}{|S|} \epsilon.$$

We can similarly show that $\mathbf{B} \in \mathfrak{I}(\mathfrak{v}, \succeq)$ and $\mathbf{B}$ $(\succeq; \lll, \preceq_\Omega)$-dominates $\mathbf{A}$ via $N$.

Therefore we conclude that $\mathbf{A} \in \mathfrak{I}(\mathfrak{v}, \succeq) \setminus \mathfrak{DC}(\mathfrak{v}, \succeq; \lll, \preceq_\Omega)$, which proves $\mathfrak{DC}(\mathfrak{v}, \succeq; \lll, \preceq_\Omega) \subseteq \mathfrak{C}(\mathfrak{v}, \succeq; \prec_\Omega)$.

We can similarly obtain the inclusion $\mathfrak{DC}(\mathfrak{v}, \succeq; \lll, \preceq_\Omega) \subseteq \mathfrak{C}(\mathfrak{v}, \succeq; \prec)$ using the above argument without considering $\omega_1$ and $\omega_2$. This completes the proof. $\square$

Suppose that the cooperative game $(N, \mathfrak{v})$ with interval-valued payoff is $\succsim$-superadditive. Then we have

$$\mathfrak{v}(N) \succsim \mathfrak{v}(S) \oplus \left( \bigoplus_{i \in N \setminus S} \mathfrak{v}(\{i\}) \right)$$

for each $S \subseteq N$ with $S \neq N$.

**Theorem 2.** *Let $(N, \mathfrak{v})$ be a cooperative game with an interval-valued payoff. Suppose that*

$$\mathfrak{v}(N) \succsim \mathfrak{v}(S) \oplus \left( \bigoplus_{i \in N \setminus S} \mathfrak{v}(\{i\}) \right) \tag{6}$$

*for each $S \subseteq N$ with $S \neq N$. Then*

$$\mathfrak{DC}_H(\mathfrak{v}, \succeq; \prec, \preceq) = \mathfrak{C}_H(\mathfrak{v}, \succeq; \prec) \text{ and } \mathfrak{DC}_H(\mathfrak{v}, \succeq; \prec, \preceq_\Omega) = \mathfrak{C}_H(\mathfrak{v}, \succeq; \prec_\Omega).$$

**Proof.** For proving the inclusion $\mathfrak{DC}_H(\mathfrak{v}, \succeq; \prec, \preceq_\Omega) \subseteq \mathfrak{C}_H(\mathfrak{v}, \succeq; \prec_\Omega)$, we want to show that

$$\mathbf{A} \in \mathfrak{I}(\mathfrak{v}, \succeq) \setminus \mathfrak{C}_H(\mathfrak{v}, \succeq; \prec_\Omega) = \widehat{\mathfrak{C}}_H(\mathfrak{v}, \succeq; \prec_\Omega) \text{ implies } \mathbf{A} \in \mathfrak{I}(\mathfrak{v}, \succeq) \setminus \mathfrak{DC}_H(\mathfrak{v}, \succeq; \prec, \preceq_\Omega).$$

For $\mathbf{A} \in \widehat{\mathfrak{C}}_H(\mathfrak{v}, \succeq; \prec_\Omega)$, by the definition of anti-core, there exists $S \subseteq N$ such that

$$\bigoplus_{i \in S} A_i \oplus \omega_1 \prec \mathfrak{v}(S) \oplus \omega_1 \text{ for some } \omega_1, \omega_2 \in \Omega \tag{7}$$

and the Hukuhara difference

$$\epsilon \equiv (\mathfrak{v}(S) \oplus \omega_2) \ominus_H \left( \bigoplus_{i \in S} A_i \oplus \omega_1 \right) \text{ exists .} \tag{8}$$

By the definition of Hukuhara difference, we see that

$$\epsilon^L = (\mathfrak{v}(S) \oplus \omega_2)^L - \left( \bigoplus_{i \in S} A_i \oplus \omega_1 \right)^L \text{ and } \epsilon^U = (\mathfrak{v}(S) \oplus \omega_2)^U - \left( \bigoplus_{i \in S} A_i \oplus \omega_1 \right)^U.$$

Therefore, from (7), we obtain $\epsilon^U \geq \epsilon^L \geq 0$ with $\epsilon^U > 0$ or $\epsilon^L > 0$. We also have

$$\epsilon \oplus \left( \bigoplus_{i \in S} A_i \oplus \omega_1 \right) = \mathfrak{v}(S) \oplus \omega_2. \tag{9}$$

We are going to find an imputation **B** such that **B** $(\succeq; \prec, \preceq_\Omega)$-H-dominates **A** via $S$. Suppose that $S \neq N$, i.e., $N \setminus S \neq \emptyset$, we define

$$B_i = \begin{cases} A_i \oplus \frac{1}{|S|}\epsilon, & \text{if } i \in S \\ \mathfrak{v}(\{i\}) \oplus \left( \frac{1}{|N \setminus S|} \left[ \mathfrak{v}(N) \ominus \left( \bigoplus_{i \in N \setminus S} \mathfrak{v}(\{i\}) \right) \ominus \mathfrak{v}(S) \right] \right), & \text{if } i \in N \setminus S. \end{cases}$$

Then, according to (9), we have

$$\bigoplus_{i \in S} B_i \oplus \omega_1 = \epsilon \oplus \left( \bigoplus_{i \in S} A_i \oplus \omega_1 \right) = \mathfrak{v}(S) \oplus \omega_2. \tag{10}$$

We also have

$$\bigoplus_{i \in N \setminus S} B_i = \bigoplus_{i \in N \setminus S} \mathfrak{v}(\{i\}) \oplus \left[ \mathfrak{v}(N) \ominus \left( \bigoplus_{i \in N \setminus S} \mathfrak{v}(\{i\}) \right) \ominus \mathfrak{v}(S) \right]. \tag{11}$$

Let

$$\omega_3 = \left( \bigoplus_{i \in N \setminus S} \mathfrak{v}(\{i\}) \right) \ominus \left( \bigoplus_{i \in N \setminus S} \mathfrak{v}(\{i\}) \right) \in \Omega \text{ and } \omega_4 = \mathfrak{v}(S) \ominus \mathfrak{v}(S) \in \Omega. \tag{12}$$

Then, from (10)–(12), we obtain

$$\bigoplus_{i \in N} B_i \oplus \omega_1 = \omega_1 \oplus \bigoplus_{i \in S} B_i \oplus \bigoplus_{i \in N \setminus S} B_i = \mathfrak{v}(S) \oplus \omega_2 \oplus \omega_3 \oplus \mathfrak{v}(N) \ominus \mathfrak{v}(S)$$

$$= \omega_2 \oplus \omega_3 \oplus \omega_4 \oplus \mathfrak{v}(N),$$

which says that

$$\bigoplus_{i \in N} B_i =_\Omega \mathfrak{v}(N)$$

by the fact of $\omega_2 \oplus \omega_3 \oplus \omega_4 \in \Omega$.

Since **A** $\in \mathfrak{I}(\mathfrak{v}, \succeq)$ and $\epsilon^U \geq \epsilon^L \geq 0$ with $\epsilon^U > 0$ or $\epsilon^L > 0$, we have

$$B_i = A_i \oplus \frac{1}{|S|}\epsilon \succ A_i \succeq \mathfrak{v}(\{i\}) \text{ for each } i \in S,$$

which also implies $B_i \succ \mathfrak{v}(\{i\})$, i.e., $B_i \succeq \mathfrak{v}(\{i\})$ for $i \in S$. From (6), we see that $B_i \succeq \mathfrak{v}(\{i\})$ for $i \in N \setminus S$.

From (10), we also have

$$\bigoplus_{i \in S} B_i \preceq_\Omega \mathfrak{v}(S).$$

Since the Hukuhara difference in (8) exists, it says that **B** $\in \mathfrak{I}(\mathfrak{v}, \succeq)$ and **B** $(\succeq; \prec, \preceq_\Omega)$-H-dominates **A** via $S$.

Suppose that $S = N$, for each $i \in N$, we define

$$B_i = A_i \oplus \frac{1}{|N|}.$$

We can similarly show that $\mathbf{B} \in \mathfrak{I}(\mathfrak{v}, \succeq)$ and $\mathbf{B}$ $(\succeq; \prec, \preceq_\Omega)$-H-dominates $\mathbf{A}$ via $N$. Therefore we conclude that $\mathbf{A} \in \mathfrak{I}(\mathfrak{v}, \succeq) \setminus \mathfrak{DC}_H(\mathfrak{v}, \succeq; \prec, \preceq_\Omega)$, which proves $\mathfrak{DC}_H(\mathfrak{v}, \succeq; \prec, \preceq_\Omega) \subseteq \mathfrak{C}_H(\mathfrak{v}, \succeq; \prec_\Omega)$.

Using Proposition 4, we obtain the desired equality.

We can similarly obtain the inclusion $\mathfrak{DC}_H(\mathfrak{v}, \succeq; \prec, \preceq) \subseteq \mathfrak{C}_H(\mathfrak{v}, \succeq; \prec)$ using the above argument without considering $\omega_1$ and $\omega_2$. Using Proposition 4 again, we obtain the desired equality. This completes the proof. $\square$

**Theorem 3.** *Let $(N, \mathfrak{v})$ be a cooperative game with an interval-valued payoff. Suppose that*

$$\mathfrak{v}(N) \succsim \mathfrak{v}(S) \oplus \left( \bigoplus_{i \in N \setminus S} \mathfrak{v}(\{i\}) \right) \tag{13}$$

*for each $S \subseteq N$ with $S \neq N$. Then*

$$\mathfrak{DC}(\mathfrak{v}, \succeq; \sqsubset, \preceq_\Omega) = \mathfrak{C}(\mathfrak{v}, \succeq; \sqsubset_\Omega) \text{ and } \mathfrak{DC}(\mathfrak{v}, \succeq; \sqsubset, \preceq) = \mathfrak{C}(\mathfrak{v}, \succeq; \sqsubset).$$

**Proof.** For proving the inclusion $\mathfrak{DC}(\mathfrak{v}, \succeq; \sqsubset, \preceq_\Omega) \subseteq \mathfrak{C}(\mathfrak{v}, \succeq; \sqsubset_\Omega)$, we want to show that

$$\mathbf{A} \in \mathfrak{I}(\mathfrak{v}, \succeq) \setminus \mathfrak{C}(\mathfrak{v}, \succeq; \sqsubset_\Omega) = \widehat{\mathfrak{C}}(\mathfrak{v}, \succeq; \sqsubset_\Omega) \text{ implies } \mathbf{A} \in \mathfrak{I}(\mathfrak{v}, \succeq) \setminus \mathfrak{DC}(\mathfrak{v}, \succeq; \sqsubset, \preceq_\Omega).$$

For $\mathbf{A} \in \widehat{\mathfrak{C}}(\mathfrak{v}, \succeq; \sqsubset_\Omega)$, by the definition of anti-core, there exists $S \subseteq N$ such that $\bigoplus_{i \in S} A_i \sqsubset_\Omega \mathfrak{v}(S)$, i.e.,

$$\bigoplus_{i \in S} A_i \oplus \omega_1 \sqsubset_\Omega \mathfrak{v}(S) \oplus \omega_2 \tag{14}$$

for some $\omega_1, \omega_2 \in \Omega$. Let

$$\epsilon = (\mathfrak{v}(S) \oplus \omega_2) \ominus \left( \bigoplus_{i \in S} A_i \oplus \omega_1 \right).$$

Using Proposition 1, we have

$$\epsilon \oplus \left( \bigoplus_{i \in S} A_i \right) =_\Omega \mathfrak{v}(S). \tag{15}$$

From (14), it follows that $\epsilon^U > 0$. We are going to find an imputation $\mathbf{B}$ such that $\mathbf{B}$ $(\succeq; \sqsubset, \preceq_\Omega)$-dominates $\mathbf{A}$ via $S$.

Suppose that $S \neq N$, i.e., $N \setminus S \neq \emptyset$, we define

$$B_i = \begin{cases} A_i \oplus \frac{1}{|S|} \epsilon, & \text{if } i \in S \\ \mathfrak{v}(\{i\}) \oplus \left( \frac{1}{|N \setminus S|} \left[ \mathfrak{v}(N) \ominus \left( \bigoplus_{i \in N \setminus S} \mathfrak{v}(\{i\}) \right) \ominus \mathfrak{v}(S) \right] \right), & \text{if } i \in N \setminus S. \end{cases}$$

Then, according to (15), we have

$$\bigoplus_{i \in S} B_i = \epsilon \oplus \left( \bigoplus_{i \in S} A_i \right) =_\Omega \mathfrak{v}(S),$$

which says that

$$\bigoplus_{i \in S} B_i \oplus \omega_3 = \mathfrak{v}(S) \oplus \omega_4 \tag{16}$$

for some $\omega_3, \omega_4 \in \Omega$. We also have

$$\bigoplus_{i \in N \setminus S} B_i = \bigoplus_{i \in N \setminus S} \mathfrak{v}(\{i\}) \oplus \left[ \mathfrak{v}(N) \ominus \left( \bigoplus_{i \in N \setminus S} \mathfrak{v}(\{i\}) \right) \ominus \mathfrak{v}(S) \right]. \tag{17}$$

Let

$$\omega_5 = \left( \bigoplus_{i \in N \setminus S} \mathfrak{v}(\{i\}) \right) \ominus \left( \bigoplus_{i \in N \setminus S} \mathfrak{v}(\{i\}) \right) \in \Omega \text{ and } \omega_6 = \mathfrak{v}(S) \ominus \mathfrak{v}(S) \in \Omega. \tag{18}$$

Then, from (16), (17) and (18), we obtain

$$\bigoplus_{i \in N} B_i \oplus \omega_3 = \omega_3 \oplus \bigoplus_{i \in S} B_i \oplus \bigoplus_{i \in N \setminus S} B_i = \omega_4 \oplus \mathfrak{v}(S) \oplus \omega_5 \oplus \mathfrak{v}(N) \ominus \mathfrak{v}(S)$$

$$= \omega_4 \oplus \omega_5 \oplus \omega_6 \oplus \mathfrak{v}(N),$$

which says that

$$\bigoplus_{i \in N} B_i =_\Omega \mathfrak{v}(N)$$

by the fact of $\omega_4 \oplus \omega_5 \oplus \omega_6 \in \Omega$.

Since $\mathbf{A} \in \mathfrak{I}(\mathfrak{v}, \succeq)$ and $\epsilon^U > 0$, we have

$$B_i = A_i \oplus \frac{1}{|S|} \epsilon \sqsupset A_i \succeq \mathfrak{v}(\{i\}) \text{ for each } i \in S,$$

which also implies $B_i \succeq \mathfrak{v}(\{i\})$ for $i \in S$. From (13), we see that $B_i \succeq \mathfrak{v}(\{i\})$ for $i \in N \setminus S$.

From (16), we also have

$$\bigoplus_{i \in S} B_i \preceq_\Omega \mathfrak{v}(S).$$

This shows that $\mathbf{B} \in \mathfrak{I}(\mathfrak{v}, \succeq)$ and $\mathbf{B}$ $(\succeq; \sqsubset, \preceq_\Omega)$-dominates $\mathbf{A}$ via $S$.

Suppose that $S = N$, for each $i \in N$, we define

$$B_i = A_i \oplus \frac{1}{|N|} \epsilon.$$

We can similarly show that $\mathbf{B} \in \mathfrak{I}(\mathfrak{v}, \succeq)$ and $\mathbf{B}$ $(\succeq; \sqsubset, \preceq_\Omega)$-dominates $\mathbf{A}$ via $N$. Therefore we conclude that $\mathbf{A} \in \mathfrak{I}(\mathfrak{v}, \succeq) \setminus \mathfrak{DC}(\mathfrak{v}, \succeq; \sqsubset, \preceq_\Omega)$, which proves $\mathfrak{DC}(\mathfrak{v}, \succeq; \sqsubset, \preceq_\Omega) \subseteq \mathfrak{C}(\mathfrak{v}, \succeq; \sqsubset_\Omega)$.

Using Proposition 3, we obtain the desired equality.

We can similarly obtain the inclusion $\mathfrak{DC}(\mathfrak{v}, \succeq; \sqsubset, \preceq) \subseteq \mathfrak{C}(\mathfrak{v}, \succeq; \sqsubset)$ using the above argument without considering $\omega_1$ and $\omega_2$. Using Proposition 3 again, we obtain the desired equality. This completes the proof. □

Based on the different binary relations, we can similarly establish the other types of dominance cores (resp. H-dominance cores) that is contained in the cores (resp. H-cores) like Theorem 1, or the other types of dominance cores (resp. H-dominance cores) that is equal to the cores (resp. H-cores) like Theorems 2 and 3.

We omit the details.

**Example 6.** *Continued from Examples 4 and 5, let*

$$\tilde{u}(S) = \mathfrak{v}(S) \oplus \left( \bigoplus_{i \in N \setminus S} \mathfrak{v}(\{i\}) \right).$$

*Suppose that*

$$\left(B \otimes \tilde{1}_{|N \setminus S|}\right)_\alpha^L \geq (\tilde{u}(S))_\alpha^U - (\tilde{u}(S))_\alpha^L$$

*for all $\alpha \in [0,1]$. Then we can show that*

$$\mathfrak{v}(N) \succsim \mathfrak{v}(S) \oplus \left(\bigoplus_{i \in N \setminus S} \mathfrak{v}(\{i\})\right).$$

*Using the above theorems, we have the following results:*

- $\mathfrak{DC}(\mathfrak{v}, \succeq; \ll, \preceq_\Omega) \subseteq \mathfrak{C}(\mathfrak{v}, \succeq; \lessdot_\Omega)$ *and* $\mathfrak{DC}(\mathfrak{v}, \succeq; \ll, \preceq_\Omega) \subseteq \mathfrak{C}(\mathfrak{v}, \succeq; \lessdot)$;
- $\mathfrak{DC}_H(\mathfrak{v}, \succeq; \prec, \preceq) = \mathfrak{C}_H(\mathfrak{v}, \succeq; \prec)$ *and* $\mathfrak{DC}_H(\mathfrak{v}, \succeq; \prec, \preceq_\Omega) = \mathfrak{C}_H(\mathfrak{v}, \succeq; \prec_\Omega)$;
- $\mathfrak{DC}(\mathfrak{v}, \succeq; \sqsubset, \preceq_\Omega) = \mathfrak{C}(\mathfrak{v}, \succeq; \sqsubset_\Omega)$ *and* $\mathfrak{DC}(\mathfrak{v}, \succeq; \sqsubset, \preceq) = \mathfrak{C}(\mathfrak{v}, \succeq; \sqsubset)$.

## 8. The Relations between Pre-Cores and Dominance Pre-Cores

Now we study the relations between pre-cores (resp. H-pre-cores) and dominance pre-cores (resp. H-dominance pre-cores). We shall see that the extra inequalities are not needed. Therefore the proofs are different.

**Proposition 5.** *Given a cooperative game with an interval-valued payoff $(N, \mathfrak{v})$, under the set of imputation $\mathfrak{I}^\circ(\mathfrak{v}, \succeq)$, we have the following inclusions:*

- $\mathfrak{C}^\circ(\mathfrak{v}, \succeq; \lessdot_\Omega) \subseteq \mathfrak{DC}^\circ(\mathfrak{v}, \succeq; \sqsubset, \precsim_\Omega)$ *and* $\mathfrak{C}^\circ(\mathfrak{v}, \succeq; \lessdot) \subseteq \mathfrak{DC}^\circ(\mathfrak{v}, \succeq; \sqsubset, \precsim)$;
- $\mathfrak{C}^\circ(\mathfrak{v}, \succeq; \sqsubset_\Omega) \subseteq \mathfrak{DC}^\circ(\mathfrak{v}, \succeq; \sqsubset, \preccurlyeq_\Omega)$ *and* $\mathfrak{C}^\circ(\mathfrak{v}, \succeq; \sqsubset) \subseteq \mathfrak{DC}^\circ(\mathfrak{v}, \succeq; \sqsubset, \preccurlyeq)$;
- $\mathfrak{C}^\circ(\mathfrak{v}, \succeq; \sqsubset_\Omega) \subseteq \mathfrak{DC}^\circ(\mathfrak{v}, \succeq; \sqsubset, \preceq_\Omega)$ *and* $\mathfrak{C}^\circ(\mathfrak{v}, \succeq; \sqsubset) \subseteq \mathfrak{DC}^\circ(\mathfrak{v}, \succeq; \sqsubset, \preceq)$;
- $\mathfrak{C}^\circ(\mathfrak{v}, \succeq; \prec_\Omega) \subseteq \mathfrak{DC}^\circ(\mathfrak{v}, \succeq; \prec, \preceq_\Omega)$ *and* $\mathfrak{C}^\circ(\mathfrak{v}, \succeq; \prec) \subseteq \mathfrak{DC}^\circ(\mathfrak{v}, \succeq; \prec, \preceq)$;
- $\mathfrak{C}^\circ(\mathfrak{v}, \succeq; \sqsubset_\Omega) \subseteq \mathfrak{DC}^\circ(\mathfrak{v}, \succeq; \prec, \preccurlyeq_\Omega)$ *and* $\mathfrak{C}^\circ(\mathfrak{v}, \succeq; \sqsubset) \subseteq \mathfrak{DC}^\circ(\mathfrak{v}, \succeq; \prec, \preccurlyeq)$;
- $\mathfrak{C}^\circ(\mathfrak{v}, \succeq; \lessdot_\Omega) \subseteq \mathfrak{DC}^\circ(\mathfrak{v}, \succeq; \prec, \precsim_\Omega)$ *and* $\mathfrak{C}^\circ(\mathfrak{v}, \succeq; \lessdot) \subseteq \mathfrak{DC}^\circ(\mathfrak{v}, \succeq; \prec, \precsim)$;
- $\mathfrak{C}^\circ(\mathfrak{v}, \succeq; \ll_\Omega) \subseteq \mathfrak{DC}^\circ(\mathfrak{v}, \succeq; \ll, \preceq_\Omega)$ *and* $\mathfrak{C}^\circ(\mathfrak{v}, \succeq; \ll) \subseteq \mathfrak{DC}^\circ(\mathfrak{v}, \succeq; \ll, \preceq)$;
- $\mathfrak{C}^\circ(\mathfrak{v}, \succeq; \lessdot_\Omega) \subseteq \mathfrak{DC}^\circ(\mathfrak{v}, \succeq; \ll, \precsim_\Omega)$ *and* $\mathfrak{C}^\circ(\mathfrak{v}, \succeq; \lessdot) \subseteq \mathfrak{DC}^\circ(\mathfrak{v}, \succeq; \ll, \precsim)$;
- $\mathfrak{C}^\circ(\mathfrak{v}, \succeq; \sqsubset_\Omega) \subseteq \mathfrak{DC}^\circ(\mathfrak{v}, \succeq; \ll, \preccurlyeq_\Omega)$ *and* $\mathfrak{C}^\circ(\mathfrak{v}, \succeq; \sqsubset) \subseteq \mathfrak{DC}^\circ(\mathfrak{v}, \succeq; \ll, \preccurlyeq)$;
- $\mathfrak{C}^\circ(\mathfrak{v}, \succeq; \prec_\Omega) \subseteq \mathfrak{DC}^\circ(\mathfrak{v}, \succeq; \lessdot, \preceq_\Omega)$ *and* $\mathfrak{C}^\circ(\mathfrak{v}, \succeq; \prec) \subseteq \mathfrak{DC}^\circ(\mathfrak{v}, \succeq; \lessdot, \preceq)$;
- $\mathfrak{C}^\circ(\mathfrak{v}, \succeq; \sqsubset_\Omega) \subseteq \mathfrak{DC}^\circ(\mathfrak{v}, \succeq; \lessdot, \preccurlyeq_\Omega)$ *and* $\mathfrak{C}^\circ(\mathfrak{v}, \succeq; \sqsubset) \subseteq \mathfrak{DC}^\circ(\mathfrak{v}, \succeq; \lessdot, \preccurlyeq)$;
- $\mathfrak{C}^\circ(\mathfrak{v}, \succeq; \lessdot_\Omega) \subseteq \mathfrak{DC}^\circ(\mathfrak{v}, \succeq; \lessdot, \precsim_\Omega)$ *and* $\mathfrak{C}^\circ(\mathfrak{v}, \succeq; \lessdot) \subseteq \mathfrak{DC}^\circ(\mathfrak{v}, \succeq; \lessdot, \precsim)$;

**Proof.** Since the individual rationality $A_i \succeq \mathfrak{v}(\{i\})$ is not considered in $\mathfrak{I}^\circ(\mathfrak{v}, \succeq)$ for $i \in N$, it is clear that the proof of Proposition 3 is still valid. This completes the proof. □

**Proposition 6.** *Given a cooperative game with interval-valued payoff $(N, \mathfrak{v})$, under the set of imputation $\mathfrak{I}^\circ(\mathfrak{v}, \succeq)$, we have the following inclusions:*

- $\mathfrak{C}^\circ_H(\mathfrak{v}, \succeq; \lessdot_\Omega) \subseteq \mathfrak{DC}^\circ_H(\mathfrak{v}, \succeq; \sqsubset, \precsim_\Omega)$ *and* $\mathfrak{C}^\circ_H(\mathfrak{v}, \succeq; \lessdot) \subseteq \mathfrak{DC}^\circ_H(\mathfrak{v}, \succeq; \sqsubset, \precsim)$;
- $\mathfrak{C}^\circ_H(\mathfrak{v}, \succeq; \sqsubset_\Omega) \subseteq \mathfrak{DC}^\circ_H(\mathfrak{v}, \succeq; \sqsubset, \preccurlyeq_\Omega)$ *and* $\mathfrak{C}^\circ_H(\mathfrak{v}, \succeq; \sqsubset) \subseteq \mathfrak{DC}^\circ_H(\mathfrak{v}, \succeq; \sqsubset, \preccurlyeq)$;
- $\mathfrak{C}^\circ_H(\mathfrak{v}, \succeq; \sqsubset_\Omega) \subseteq \mathfrak{DC}^\circ_H(\mathfrak{v}, \succeq; \sqsubset, \preceq_\Omega)$ *and* $\mathfrak{C}^\circ_H(\mathfrak{v}, \succeq; \sqsubset) \subseteq \mathfrak{DC}^\circ_H(\mathfrak{v}, \succeq; \sqsubset, \preceq)$;
- $\mathfrak{C}^\circ_H(\mathfrak{v}, \succeq; \prec_\Omega) \subseteq \mathfrak{DC}^\circ_H(\mathfrak{v}, \succeq; \prec, \preceq_\Omega)$ *and* $\mathfrak{C}^\circ_H(\mathfrak{v}, \succeq; \prec) \subseteq \mathfrak{DC}^\circ_H(\mathfrak{v}, \succeq; \prec, \preceq)$;
- $\mathfrak{C}^\circ_H(\mathfrak{v}, \succeq; \sqsubset_\Omega) \subseteq \mathfrak{DC}^\circ_H(\mathfrak{v}, \succeq; \prec, \preccurlyeq_\Omega)$ *and* $\mathfrak{C}^\circ_H(\mathfrak{v}, \succeq; \sqsubset) \subseteq \mathfrak{DC}^\circ_H(\mathfrak{v}, \succeq; \prec, \preccurlyeq)$;
- $\mathfrak{C}^\circ_H(\mathfrak{v}, \succeq; \lessdot_\Omega) \subseteq \mathfrak{DC}^\circ_H(\mathfrak{v}, \succeq; \prec, \precsim_\Omega)$ *and* $\mathfrak{C}^\circ_H(\mathfrak{v}, \succeq; \lessdot) \subseteq \mathfrak{DC}^\circ_H(\mathfrak{v}, \succeq; \prec, \precsim)$;
- $\mathfrak{C}^\circ_H(\mathfrak{v}, \succeq; \ll_\Omega) \subseteq \mathfrak{DC}^\circ_H(\mathfrak{v}, \succeq; \ll, \preceq_\Omega)$ *and* $\mathfrak{C}^\circ_H(\mathfrak{v}, \succeq; \ll) \subseteq \mathfrak{DC}^\circ_H(\mathfrak{v}, \succeq; \ll, \preceq)$;
- $\mathfrak{C}^\circ_H(\mathfrak{v}, \succeq; \lessdot_\Omega) \subseteq \mathfrak{DC}^\circ_H(\mathfrak{v}, \succeq; \ll, \precsim_\Omega)$ *and* $\mathfrak{C}^\circ_H(\mathfrak{v}, \succeq; \lessdot) \subseteq \mathfrak{DC}^\circ_H(\mathfrak{v}, \succeq; \ll, \precsim)$;
- $\mathfrak{C}^\circ_H(\mathfrak{v}, \succeq; \sqsubset_\Omega) \subseteq \mathfrak{DC}^\circ_H(\mathfrak{v}, \succeq; \ll, \preccurlyeq_\Omega)$ *and* $\mathfrak{C}^\circ_H(\mathfrak{v}, \succeq; \sqsubset) \subseteq \mathfrak{DC}^\circ_H(\mathfrak{v}, \succeq; \ll, \preccurlyeq)$;

- $\mathfrak{C}_H^\circ(\mathfrak{v}, \succeq; \prec_\Omega) \subseteq \mathfrak{DC}_H^\circ(\mathfrak{v}, \succeq; \lessdot, \preceq_\Omega)$ *and* $\mathfrak{C}_H^\circ(\mathfrak{v}, \succeq; \prec) \subseteq \mathfrak{DC}_H^\circ(\mathfrak{v}, \succeq; \lessdot, \preceq)$;
- $\mathfrak{C}_H^\circ(\mathfrak{v}, \succeq; \sqsubset_\Omega) \subseteq \mathfrak{DC}_H^\circ(\mathfrak{v}, \succeq; \lessdot, \precsim_\Omega)$ *and* $\mathfrak{C}_H^\circ(\mathfrak{v}, \succeq; \sqsubset) \subseteq \mathfrak{DC}_H^\circ(\mathfrak{v}, \succeq; \lessdot, \precsim)$;
- $\mathfrak{C}_H^\circ(\mathfrak{v}, \succeq; \lessdot_\Omega) \subseteq \mathfrak{DC}_H^\circ(\mathfrak{v}, \succeq; \lessdot, \precsim_\Omega)$ *and* $\mathfrak{C}_H^\circ(\mathfrak{v}, \succeq; \lessdot) \subseteq \mathfrak{DC}_H^\circ(\mathfrak{v}, \succeq; \lessdot, \precsim)$;

**Proof.** Since the individual rationality $A_i \succeq \mathfrak{v}(\{i\})$ is not considered in $\mathfrak{I}^\circ(\mathfrak{v}, \succeq)$ for $i \in N$, it is clear that the proof of Proposition 4 is still valid. This completes the proof. $\square$

The inclusions in Propositions 5 and 6 are under the set of imputation $\mathfrak{I}^\circ(\mathfrak{v}, \succeq)$. We can obtain the similar inclusions under the different sets of imputation. We omit the details.

**Theorem 4.** *Let* $(N, \mathfrak{v})$ *be a cooperative game with an interval-valued payoff. Then*

$$\mathfrak{DC}^\circ(\mathfrak{v}, \succeq; \lll, \preceq_\Omega) \subseteq \mathfrak{C}^\circ(\mathfrak{v}, \succeq; \lessdot_\Omega) \text{ and } \mathfrak{DC}^\circ(\mathfrak{v}, \succeq; \lll, \preceq_\Omega) \subseteq \mathfrak{C}^\circ(\mathfrak{v}, \succeq; \lessdot).$$

**Proof.** For proving the inclusion $\mathfrak{DC}^\circ(\mathfrak{v}, \succeq; \lll, \preceq_\Omega) \subseteq \mathfrak{C}^\circ(\mathfrak{v}, \succeq; \lessdot_\Omega)$, we want to show that

$$\mathbf{A} \in \mathfrak{I}^\circ(\mathfrak{v}, \succeq) \setminus \mathfrak{C}^\circ(\mathfrak{v}, \succeq; \lessdot_\Omega) = \widehat{\mathfrak{C}}^*(\mathfrak{v}, \succeq; \lessdot_\Omega) \text{ implies } \mathbf{A} \in \mathfrak{I}^\circ(\mathfrak{v}, \succeq) \setminus \mathfrak{DC}^\circ(\mathfrak{v}, \succeq; \lll, \preceq_\Omega).$$

For $\mathbf{A} \in \widehat{\mathfrak{C}}^*(\mathfrak{v}, \succeq; \lessdot_\Omega)$, by the definition of anti-pre-core, there exists $S \subseteq N$ such that $\bigoplus_{i \in S} A_i \lessdot_\Omega \mathfrak{v}(S)$, i.e., $\bigoplus_{i \in S} A_i \oplus \omega_1 < \mathfrak{v}(S) \oplus \omega_2$ for some $\omega_1, \omega_2 \in \Omega$. Let

$$\epsilon = (\mathfrak{v}(S) \oplus \omega_2) \ominus \left( \bigoplus_{i \in S} A_i \oplus \omega_1 \right).$$

Since $\bigoplus_{i \in S} A_i \oplus \omega_1 < \mathfrak{v}(S) \oplus \omega_2$, it follows that $\epsilon^U \geq \epsilon^L > 0$. Using part (ii) of Proposition 1, we obtain

$$\epsilon \oplus \left( \bigoplus_{i \in S} A_i \right) =_\Omega \mathfrak{v}(S), \tag{19}$$

We are going to find a pre-imputation $\mathbf{B}$ such that $\mathbf{B}$ $(\succeq; \lll, \preceq_\Omega)$-dominates $\mathbf{A}$ via $S$. Suppose that $S \neq N$, i.e., $N \setminus S \neq \varnothing$, we define

$$B_i = \begin{cases} A_i \oplus \frac{1}{|S|} \epsilon, & \text{if } i \in S \\ \frac{1}{|N \setminus S|} [\mathfrak{v}(N) \ominus \mathfrak{v}(S)], & \text{if } i \in N \setminus S. \end{cases}$$

Then, according to (19), we have

$$\bigoplus_{i \in S} B_i = \left( \bigoplus_{i \in S} A_i \right) \oplus \epsilon =_\Omega \mathfrak{v}(S),$$

which says that

$$\bigoplus_{i \in S} B_i \oplus \omega_3 = \mathfrak{v}(S) \oplus \omega_4 \tag{20}$$

for some $\omega_3, \omega_4 \in \Omega$. We also have

$$\bigoplus_{i \in N \setminus S} B_i = \mathfrak{v}(N) \ominus \mathfrak{v}(S). \tag{21}$$

Let $\omega_5 = \mathfrak{v}(S) \ominus \mathfrak{v}(S) \in \Omega$.

Then, from (20) and (21), we obtain

$$\bigoplus_{i \in N} B_i \oplus \omega_3 = \omega_3 \oplus \bigoplus_{i \in S} B_i \oplus \bigoplus_{i \in N \setminus S} B_i = \omega_4 \oplus \mathfrak{v}(S) \oplus \mathfrak{v}(N) \ominus \mathfrak{v}(S)$$

$$= \omega_4 \oplus \omega_5 \oplus \mathfrak{v}(N),$$

which says that

$$\bigoplus_{i \in N} B_i =_\Omega \mathfrak{v}(N)$$

by the fact of $\omega_4 \oplus \omega_5 \in \Omega$.

Since $\epsilon^U \geq \epsilon^L > 0$, we have

$$B_i = A_i \oplus \left( \frac{1}{|S|} \otimes \epsilon \right) \ggcurly A_i \text{ for each } i \in S,$$

From (20), we also have

$$\bigoplus_{i \in S} B_i \preceq_\Omega \mathfrak{v}(S).$$

This shows that $\mathbf{B} \in \mathfrak{I}^\circ(\mathfrak{v}, \succeq)$ and $\mathbf{B}$ $(\succeq; \lllcurly, \preceq_\Omega)$-dominates $\mathbf{A}$ via $S$.

Suppose that $S = N$, for each $i \in N$, we define

$$B_i = A_i \oplus \left( \frac{1}{|S|} \otimes \epsilon \right).$$

We can similarly show that $\mathbf{B} \in \mathfrak{I}^\circ(\mathfrak{v}, \succeq)$ and $\mathbf{B}$ $(\succeq; \lllcurly, \preceq_\Omega)$-dominates $\mathbf{A}$ via $N$.

Therefore we conclude that $\mathbf{A} \in \mathfrak{I}^\circ(\mathfrak{v}, \succeq) \setminus \mathfrak{DC}^\circ(\mathfrak{v}, \succeq; \lllcurly, \preceq_\Omega)$, which proves $\mathfrak{DC}^\circ(\mathfrak{v}, \succeq; \lllcurly, \preceq_\Omega) \subseteq \mathfrak{C}^\circ(\mathfrak{v}, \succeq; \prec_\Omega)$.

We can similarly obtain the inclusion $\mathfrak{DC}^\circ(\mathfrak{v}, \succeq; \lllcurly, \preceq_\Omega) \subseteq \mathfrak{C}^\circ(\mathfrak{v}, \succeq; \prec)$ using the above argument without considering $\omega_1$ and $\omega_2$. This completes the proof. □

**Theorem 5.** *Let $(N, \mathfrak{v})$ be a cooperative game with an interval-valued payoff. Then*

$$\mathfrak{DC}_H^\circ(\mathfrak{v}, \succeq; \prec, \preceq) = \mathfrak{C}_H^\circ(\mathfrak{v}, \succeq; \prec) \text{ and } \mathfrak{DC}_H^\circ(\mathfrak{v}, \succeq; \prec, \preceq_\Omega) = \mathfrak{C}_H^\circ(\mathfrak{v}, \succeq; \prec_\Omega).$$

**Proof.** For proving the inclusion $\mathfrak{DC}_H^\circ(\mathfrak{v}, \succeq; \prec, \preceq_\Omega) \subseteq \mathfrak{C}_H^\circ(\mathfrak{v}, \succeq; \prec_\Omega)$, we want to show that

$$\mathbf{A} \in \mathfrak{I}^\circ(\mathfrak{v}, \succeq) \setminus \mathfrak{C}_H^\circ(\mathfrak{v}, \succeq; \prec_\Omega) = \widehat{\mathfrak{C}^\circ}_H(\mathfrak{v}, \succeq; \prec_\Omega) \text{ implies } \mathbf{A} \in \mathfrak{I}^\circ(\mathfrak{v}, \succeq) \setminus \mathfrak{DC}_H^\circ(\mathfrak{v}, \succeq; \prec, \preceq_\Omega).$$

For $\mathbf{A} \in \widehat{\mathfrak{C}^\circ}_H(\mathfrak{v}, \succeq; \prec_\Omega)$, by the definition of anti-pre-core, there exists $S \subseteq N$ such that

$$\bigoplus_{i \in S} A_i \oplus \omega_1 \prec \mathfrak{v}(S) \oplus \omega_1 \text{ for some } \omega_1, \omega_2 \in \Omega \tag{22}$$

and the Hukuhara difference

$$\epsilon \equiv (\mathfrak{v}(S) \oplus \omega_2) \ominus_H \left( \bigoplus_{i \in S} A_i \oplus \omega_1 \right) \text{ exists .} \tag{23}$$

By the definition of Hukuhara difference, we see that

$$\epsilon^L = (\mathfrak{v}(S) \oplus \omega_2)^L - \left( \bigoplus_{i \in S} A_i \oplus \omega_1 \right)^L \text{ and } \epsilon^U = (\mathfrak{v}(S) \oplus \omega_2)^U - \left( \bigoplus_{i \in S} A_i \oplus \omega_1 \right)^U.$$

Therefore, from (22), we obtain $\epsilon^U \geq \epsilon^L \geq 0$ with $\epsilon^U > 0$ or $\epsilon^L > 0$. We also have

$$\epsilon \oplus \left( \bigoplus_{i \in S} A_i \oplus \omega_1 \right) = \mathfrak{v}(S) \oplus \omega_2. \tag{24}$$

We are going to find an imputation **B** such that **B** $(\succeq; \prec, \preceq_\Omega)$-H-dominates **A** via $S$. Suppose that $S \neq N$, i.e., $N \setminus S \neq \varnothing$, we define

$$B_i = \begin{cases} A_i \oplus \frac{1}{|S|}\epsilon, & \text{if } i \in S \\ \frac{1}{|N \setminus S|} [\mathfrak{v}(N) \ominus \mathfrak{v}(S)], & \text{if } i \in N \setminus S. \end{cases}$$

Then, according to (24), we have

$$\bigoplus_{i \in S} B_i \oplus \omega_1 = \epsilon \oplus \left( \bigoplus_{i \in S} A_i \oplus \omega_1 \right) = \mathfrak{v}(S) \oplus \omega_2. \tag{25}$$

We also have

$$\bigoplus_{i \in N \setminus S} B_i = \mathfrak{v}(N) \ominus \mathfrak{v}(S). \tag{26}$$

Let $\omega_3 = \mathfrak{v}(S) \ominus \mathfrak{v}(S) \in \Omega$.
Then, from (25) and (26), we obtain

$$\bigoplus_{i \in N} B_i \oplus \omega_1 = \omega_1 \oplus \bigoplus_{i \in S} B_i \oplus \bigoplus_{i \in N \setminus S} B_i = \mathfrak{v}(S) \oplus \omega_2 \oplus \mathfrak{v}(N) \ominus \mathfrak{v}(S) = \omega_2 \oplus \omega_3 \oplus \mathfrak{v}(N),$$

which says that

$$\bigoplus_{i \in N} B_i =_\Omega \mathfrak{v}(N)$$

by the fact of $\omega_2 \oplus \omega_3 \in \Omega$.

Since $\mathbf{A} \in \mathfrak{I}^\circ(\mathfrak{v}, \succeq)$ and $\epsilon^U \geq \epsilon^L \geq 0$ with $\epsilon^U > 0$ or $\epsilon^L > 0$, we have

$$B_i = A_i \oplus \frac{1}{|S|}\epsilon \succ A_i \text{ for each } i \in S.$$

From (25), we also have

$$\bigoplus_{i \in S} B_i \preceq_\Omega \mathfrak{v}(S).$$

Since the Hukuhara difference in (23) exists, it says that $\mathbf{B} \in \mathfrak{I}^\circ(\mathfrak{v}, \succeq)$ and $\mathbf{B}$ $(\succeq; \prec, \preceq_\Omega)$-H-dominates **A** via $S$.
Suppose that $S = N$, for each $i \in N$, we define

$$B_i = A_i \oplus \frac{1}{|N|}.$$

We can similarly show that $\mathbf{B} \in \mathfrak{I}^\circ(\mathfrak{v}, \succeq)$ and $\mathbf{B}$ $(\succeq; \prec, \preceq_\Omega)$-H-dominates **A** via $N$. Therefore we conclude that $\mathbf{A} \in \mathfrak{I}^\circ(\mathfrak{v}, \succeq) \setminus \mathfrak{DC}_H^\circ(\mathfrak{v}, \succeq; \prec, \preceq_\Omega)$, which proves $\mathfrak{DC}_H^\circ(\mathfrak{v}, \succeq; \prec, \preceq_\Omega) \subseteq \mathfrak{C}_H^\circ(\mathfrak{v}, \succeq; \prec_\Omega)$.

Using Proposition 6, we obtain the desired equality.

We can similarly obtain the inclusion $\mathfrak{DC}_H^\circ(\mathfrak{v}, \succeq; \prec, \preceq) \subseteq \mathfrak{C}_H^\circ(\mathfrak{v}, \succeq; \prec)$ using the above argument without considering $\omega_1$ and $\omega_2$. Using Proposition 6 again, we obtain the desired equality. This completes the proof. $\square$

**Theorem 6.** *Let $(N, \mathfrak{v})$ be a cooperative game with an interval-valued payoff. Then*

$$\mathfrak{D}\mathfrak{C}^\circ(\mathfrak{v}, \succeq; \sqsubset, \preceq_\Omega) = \mathfrak{C}^\circ(\mathfrak{v}, \succeq; \sqsubset_\Omega) \text{ and } \mathfrak{D}\mathfrak{C}^\circ(\mathfrak{v}, \succeq; \sqsubset, \preceq) = \mathfrak{C}^\circ(\mathfrak{v}, \succeq; \sqsubset).$$

**Proof.** For proving the inclusion $\mathfrak{D}\mathfrak{C}^\circ(\mathfrak{v}, \succeq; \sqsubset, \preceq_\Omega) \subseteq \mathfrak{C}^\circ(\mathfrak{v}, \succeq; \sqsubset_\Omega)$, we want to show that

$$\mathbf{A} \in \mathfrak{I}^\circ(\mathfrak{v}, \succeq) \setminus \mathfrak{C}^\circ(\mathfrak{v}, \succeq; \sqsubset_\Omega) = \widehat{\mathfrak{C}^\circ}(\mathfrak{v}, \succeq; \sqsubset_\Omega) \text{ implies } \mathbf{A} \in \mathfrak{I}^\circ(\mathfrak{v}, \succeq) \setminus \mathfrak{D}\mathfrak{C}^\circ(\mathfrak{v}, \succeq; \sqsubset, \preceq_\Omega).$$

For $\mathbf{A} \in \widehat{\mathfrak{C}^\circ}(\mathfrak{v}, \succeq; \sqsubset_\Omega)$, by the definition of anti-pre-core, there exists $S \subseteq N$ such that $\bigoplus_{i\in S} A_i \sqsubset_\Omega \mathfrak{v}(S)$, i.e.,

$$\bigoplus_{i\in S} A_i \oplus \omega_1 \sqsubset_\Omega \mathfrak{v}(S) \oplus \omega_2 \tag{27}$$

for some $\omega_1, \omega_2 \in \Omega$. Let

$$\epsilon = (\mathfrak{v}(S) \oplus \omega_2) \ominus \left(\bigoplus_{i\in S} A_i \oplus \omega_1\right).$$

Using Proposition 1, we have

$$\epsilon \oplus \left(\bigoplus_{i\in S} A_i\right) =_\Omega \mathfrak{v}(S). \tag{28}$$

From (27), it follows that $\epsilon^U > 0$. We are going to find an imputation $\mathbf{B}$ such that $\mathbf{B}$ $(\succeq; \sqsubset, \preceq_\Omega)$-dominates $\mathbf{A}$ via $S$.

Suppose that $S \neq N$, i.e., $N \setminus S \neq \varnothing$, we define

$$B_i = \begin{cases} A_i \oplus \frac{1}{|S|}\epsilon, & \text{if } i \in S \\ \dfrac{1}{|N \setminus S|}[\mathfrak{v}(N) \ominus \mathfrak{v}(S)], & \text{if } i \in N \setminus S. \end{cases}$$

Then, according to (28), we have

$$\bigoplus_{i\in S} B_i = \epsilon \oplus \left(\bigoplus_{i\in S} A_i\right) =_\Omega \mathfrak{v}(S),$$

which says that

$$\bigoplus_{i\in S} B_i \oplus \omega_3 = \mathfrak{v}(S) \oplus \omega_4 \tag{29}$$

for some $\omega_3, \omega_4 \in \Omega$. We also have

$$\bigoplus_{i\in N\setminus S} B_i = \mathfrak{v}(N) \ominus \mathfrak{v}(S). \tag{30}$$

Let $\omega_5 = \mathfrak{v}(S) \ominus \mathfrak{v}(S) \in \Omega$.

Then, from (29) and (30), we obtain

$$\bigoplus_{i\in N} B_i \oplus \omega_3 = \omega_3 \oplus \bigoplus_{i\in S} B_i \oplus \bigoplus_{i\in N\setminus S} B_i = \omega_4 \oplus \mathfrak{v}(S) \oplus \mathfrak{v}(N) \ominus \mathfrak{v}(S) = \omega_4 \oplus \omega_5 \oplus \mathfrak{v}(N),$$

which says that

$$\bigoplus_{i\in N} B_i =_\Omega \mathfrak{v}(N)$$

by the fact of $\omega_4 \oplus \omega_5 \in \Omega$.

Since $\mathbf{A} \in \mathfrak{I}^{\circ}(\mathfrak{v}, \succeq)$ and $\epsilon^U > 0$, we have

$$B_i = A_i \oplus \frac{1}{|S|}\epsilon \sqsupset A_i \text{ for each } i \in S.$$

From (29), we also have

$$\bigoplus_{i \in S} B_i \preceq_{\Omega} \mathfrak{v}(S).$$

This shows that $\mathbf{B} \in \mathfrak{I}^{\circ}(\mathfrak{v}, \succeq)$ and $\mathbf{B}$ $(\succeq; \sqsubset, \preceq_{\Omega})$-dominates $\mathbf{A}$ via $S$.

Suppose that $S = N$, for each $i \in N$, we define

$$B_i = A_i \oplus \frac{1}{|N|}\epsilon.$$

We can similarly show that $\mathbf{B} \in \mathfrak{I}^{\circ}(\mathfrak{v}, \succeq)$ and $\mathbf{B}$ $(\succeq; \sqsubset, \preceq_{\Omega})$-dominates $\mathbf{A}$ via $N$. Therefore we conclude that $\mathbf{A} \in \mathfrak{I}^{\circ}(\mathfrak{v}, \succeq) \setminus \mathfrak{D}\mathfrak{C}^{\circ}(\mathfrak{v}, \succeq; \sqsubset, \preceq_{\Omega})$, which proves $\mathfrak{D}\mathfrak{C}^{\circ}(\mathfrak{v}, \succeq; \sqsubset, \preceq_{\Omega}) \subseteq \mathfrak{C}^{\circ}(\mathfrak{v}, \succeq; \sqsubset_{\Omega})$.

Using Proposition 5, we obtain the desired equality.

We can similarly obtain the inclusion $\mathfrak{D}\mathfrak{C}^{\circ}(\mathfrak{v}, \succeq; \sqsubset, \preceq) \subseteq \mathfrak{C}^{\circ}(\mathfrak{v}, \succeq; \sqsubset)$ using the above argument without considering $\omega_1$ and $\omega_2$. Using Proposition 5 again, we obtain the desired equality. This completes the proof. □

Based on the different binary relations, we can similarly establish the other types of relationships between the pre-cores (resp. H-pre-cores) and dominance pre-cores (resp. H-dominance pre-cores). We omit the details.

**Example 7.** *Continued from Examples 4 and 5, using the above theorems, we immediately have the following results:*

- $\mathfrak{D}\mathfrak{C}^{\circ}(\mathfrak{v}, \succeq; \lll, \preceq_{\Omega}) \subseteq \mathfrak{C}^{\circ}(\mathfrak{v}, \succeq; \lessdot_{\Omega})$ and $\mathfrak{D}\mathfrak{C}^{\circ}(\mathfrak{v}, \succeq; \lll, \preceq_{\Omega}) \subseteq \mathfrak{C}^{\circ}(\mathfrak{v}, \succeq; \lessdot)$;
- $\mathfrak{D}\mathfrak{C}^{\circ}_H(\mathfrak{v}, \succeq; \prec, \preceq) = \mathfrak{C}^{\circ}_H(\mathfrak{v}, \succeq; \prec)$ and $\mathfrak{D}\mathfrak{C}^{\circ}_H(\mathfrak{v}, \succeq; \prec, \preceq_{\Omega}) = \mathfrak{C}^{\circ}_H(\mathfrak{v}, \succeq; \prec_{\Omega})$;
- $\mathfrak{D}\mathfrak{C}^{\circ}(\mathfrak{v}, \succeq; \sqsubset, \preceq_{\Omega}) = \mathfrak{C}^{\circ}(\mathfrak{v}, \succeq; \sqsubset_{\Omega})$ and $\mathfrak{D}\mathfrak{C}^{\circ}(\mathfrak{v}, \succeq; \sqsubset, \preceq) = \mathfrak{C}^{\circ}(\mathfrak{v}, \succeq; \sqsubset)$.

## 9. Conclusions

In this paper, we consider the cooperative game with interval payoffs. Since the payoffs are assumed to be closed and bounded intervals in $\mathbb{R}$, the comparison between any two closed intervals are needed. Therefore many kinds of orderings and strict orderings are proposed. From Definitions 2–5, six orderings are proposed as follows

$$A \preceq B, \quad A \preccurlyeq B, \quad A \precsim B, \quad A \preceq_{\Omega} B, \quad A \preccurlyeq_{\Omega} B, \quad A \precsim_{\Omega} B, \tag{31}$$

and eight strict orderings are proposed as follows

$$A \prec B, \quad A \sqsubset B, \quad A \lessdot B, \quad A \lll B, \quad A \prec_{\Omega} B, \quad A \sqsubset_{\Omega} B, \quad A \lessdot_{\Omega} B, \quad A \lll_{\Omega} B. \tag{32}$$

In order to study the cores and dominance cores, based on the orderings in (31), six sets of imputation

$$\mathfrak{I}(\mathfrak{v}, \succeq), \quad \mathfrak{I}(\mathfrak{v}, \succeq_{\Omega}), \quad \mathfrak{I}(\mathfrak{v}, \succcurlyeq), \quad \mathfrak{I}(\mathfrak{v}, \succcurlyeq_{\Omega}), \quad \mathfrak{I}(\mathfrak{v}, \succsim) \text{ and } \mathfrak{I}(\mathfrak{v}, \succsim_{\Omega})$$

and six sets of pre-imputation

$$\mathfrak{I}^{\circ}(\mathfrak{v}, \succeq), \quad \mathfrak{I}^{\circ}(\mathfrak{v}, \succeq_{\Omega}), \quad \mathfrak{I}^{\circ}(\mathfrak{v}, \succcurlyeq), \quad \mathfrak{I}^{\circ}(\mathfrak{v}, \succcurlyeq_{\Omega}), \quad \mathfrak{I}^{\circ}(\mathfrak{v}, \succsim) \text{ and } \mathfrak{I}^{\circ}(\mathfrak{v}, \succsim_{\Omega})$$

are proposed. Under these different sets of imputation and pre-imputation, we define many kinds of anti-cores and pre-anti-cores

$$\widehat{\mathfrak{C}}(\mathfrak{v}; \mu, \nu) \text{ and } \widehat{\mathfrak{C}}^\circ(\mathfrak{v}; \mu, \nu),$$

respectively, where $\mu$ is an ordering and $\nu$ is a strict ordering given by

$$\mu \in \{\preceq, \preccurlyeq, \precsim, \preceq_\Omega, \preccurlyeq_\Omega, \precsim_\Omega\} \text{ and } \nu \in \{\prec, \sqsubset, \lessdot, \prec\!\!\prec, \prec_\Omega, \sqsubset_\Omega, \lessdot_\Omega, \prec\!\!\prec_\Omega\}.$$

Then the cores and pre-cores are defined to be the complement sets of anti-cores and pre-anti-cores, respectively, under the sets of imputations; that is, we have

$$\mathfrak{C}(\mathfrak{v}; \mu, \nu) = \mathfrak{I}(\mu) \setminus \widehat{\mathfrak{C}}(\mathfrak{v}; \mu, \nu) \text{ and } \mathfrak{C}^\circ(\mathfrak{v}; \mu, \nu) = \mathfrak{I}^\circ(\mu) \setminus \widehat{\mathfrak{C}}(\mathfrak{v}; \mu, \nu).$$

We also consider the solution concepts of dominance cores $\mathfrak{DC}(\mathfrak{v}; \mu, \nu, \lambda)$ and pre-dominance cores $\mathfrak{DC}^\circ(\mathfrak{v}; \mu, \nu, \lambda)$ where

$$\mu, \lambda \in \{\preceq, \preccurlyeq, \precsim, \preceq_\Omega, \preccurlyeq_\Omega, \precsim_\Omega\} \text{ and } \nu \in \{\prec, \sqsubset, \lessdot, \prec\!\!\prec, \prec_\Omega, \sqsubset_\Omega, \lessdot_\Omega, \prec\!\!\prec_\Omega\}.$$

Since the difference of any two closed intervals is not in a natural sense, the Hukuhara difference is considered, which leads to the different solution concepts of H-cores $\mathfrak{C}_H(\mathfrak{v}; \mu, \nu)$, H-pre-cores $\mathfrak{C}_H^\circ(\mathfrak{v}; \mu, \nu)$, H-dominance cores $\mathfrak{DC}_H(\mathfrak{v}; \mu, \nu, \lambda)$ and H-dominance pre-cores $\mathfrak{DC}_H^\circ(\mathfrak{v}; \mu, \nu, \lambda)$. The purpose of this paper is to establish the inclusion and equality relations between the cores and dominance core by referring to Theorems 1–3, and between the pre-cores and dominance pre-core by referring to Theorems 4–6.

**Funding:** This research received no external funding.

**Acknowledgments:** The author would like to thank the reviewers for providing the useful comments that definitely improve this paper.

**Conflicts of Interest:** The author declares no conflict of interest.

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
