# Peer review of "Interval-Valued Cores and Interval-Valued Dominance Cores of Cooperative Games Endowed with Interval-Valued Payoffs"

_mathematics, doi:10.3390/math6110255_

Round 1
Reviewer 1 Report
Authors must give numbers to each of equations.
Style of some of the sentences could be improved.
Introduction:
At the beginning of the paper authors must cite the author of the Game Theory and describe the principal terms used in the research:
Game theory entered economics with the publication in 1944 of the Theory of Games and Economic Behavior by John von Neumann and Oskar Morgenstern. The authors were, respectively, a Hungarian mathematician and an Austrian economist. The article discusses the creation, content and impact of that work. [von Neumann, J. and Morgenstern, O. (1944) Theory of Games and Economic Behavior, Princeton University Press, Princeton, New Jersey 08540.]
[Dimand, M.A., and R.W. Dimand. 1996. A history of game theory, Vol. I. From the Beginnings to 1945. London/New York: Routledge.]
At the foundation of the Theory of Games is the assumption that a player can evaluate concerning his utility scale every situation that can result from a play of a game [Shapley, L.S. (1951). The Value of an n-Person Game. Rand Publication RM-670. Santa Monica, CA: RAND Corporation].
Game theory concerns the behaviour of decision makers whose decisions affect each other. Its analysis is from a rational rather than a psychological or sociological viewpoint. It is indeed a sort of umbrella theory for the rational side of social science, where ‘social’ is interpreted broadly, to include human as well as non-human players (computers, animals, plants). Its methodologies apply in principle to all interactive situations, especially in economics, political science, evolutionary biology, and computer science.. [Aumann, R. J. (2017). The New Palgrave Dictionary of Economics, Palgrave Macmillan , https://link.springer.com/referenceworkentry/10.1057%2F978-1-349-95121-5_942-2]
The title ‘cooperative games’ would be better-termed games in coalitional form. The theory of games originally developed different conceptual styles, together with their associated solution concepts, namely, games in extensive form, in a strategic way, and in coalitional structure (von Neumann and Morgenstern 1944). The game is vital to form sometimes referred to like the game in standard form, while that in coalitional form also referred to like the game in characteristic style [Shubik M. (1989) Cooperative Games. In: Eatwell J., Milgate M., Newman P. (eds) Game Theory. The New Palgrave. Palgrave Macmillan, London].
Nash (1953) defined the concept of a general two-person cooperative game and a concept of a solution of such games. By using the term cooperative, he meant to imply that the players have complete freedom of communication and comprehensive information on the structure of the game. Furthermore, there should be the possibility of making enforced agreements, binding either one or both players to a particular agreement policy [Nash, J. (1953). Two-person cooperative games. Econometrica: Journal of the Econometric Society, 128-140.].
The principle of monotonicity for cooperative games states that if a game changes so that some player's contribution to all coalitions increases or stays the same, then the player's allocation should not decrease. There is a unique symmetric and efficient solution concept that is monotonic in this most general sense — the Shapley value. Monotonicity thus provides a simple characterization of the benefit without resorting to the usual “additivity” and “dummy” assumptions and lends support to the use of the value in applications where the underlying “game” is changing, e.g. in cost allocation problems [Young, H.P. (1985). Monotonic solutions of cooperative games Int J Game Theory 14: 65. https://doi.org/10.1007/BF01769885 ].
In the introduction, authors should provide some information about differences of different interval-valued cooperative game models, and show strong background why decision-makers need a new extension.
The difference between interval-valued cooperative games and crisp cooperative games is that researchers utilize intervals to express the coalitions’ values in the former rather than real numbers. Hence, in real situations, if the lower and upper bounds of all potential profits resulted [Li D-F (2016) Models And Methods of Interval-Valued Cooperative Games in Economic Management. Springer, Cham].
Presently, much research work uses intervals to estimate coalitions’ values and establishes the so-called interval-valued cooperative games (Branzei et al. 2010; Alparslan Gök et al. 2010; Mallozzi et al. 2011).
The following text I borrowed from the published paper to show how the authors must extend the literature review:
Branzei et al. (2003) considered interval-valued bankruptcy games arose from bankruptcy situations with interval claims, proposed two interval-valued Shapley-like values and studied their interrelations by using the interval arithmetic operations (Moore 1979). Mallozzi et al. (2011) introduced a core-like of cooperative games with coalitions’ values represented by fuzzy intervals (Mares 2001) and a balanced-like condition which is proven to be necessary but not sufficient to assure its non-emptiness. Han et al. (2012) proposed the interval-valued core and the interval-valued Shapley-like value of interval-valued cooperative games by defining new order relation of intervals. Branzei et al. (2011) extended the interval-valued core of interval-valued cooperative games based on the interval-valued square dominance core and interval-valued dominance core. Alparslan Gök et al. (2011) also discussed the interval-valued core, the interval-valued dominance core, and the interval-valued stable sets of interval-valued cooperative games. Alparslan Gök et al. (2009) defined the Weber set and the Shapley value for a suitable class of interval-valued cooperative games and established their relations with the interval-valued core for convex interval-valued cooperative games. Li (2016) proposed several important concepts of interval-valued solutions such as the interval-valued Shapley value, the interval-valued solidarity value as well as the interval-valued Banzhaf value and their simplified methods. Li (2016) also established an effective non-linear programming method for computing interval-valued cores of interval-valued cooperative games. However, most of the works mentioned above except Li (2016) used the partial subtraction operator or Moore’s interval subtraction (Moore 1979) which usually enlarges the uncertainty of the resulted interval.
The authors could cite the following papers (not only listed below):
Alparslan Gök SZ, Branzei O, Branzei R, Tijs S (2011) Set-valued solution concepts using interval-type payoffs for interval games. J Math Econ 47:621–626
Alparslan Gök SZ, Branzei R, Tijs S (2009) Convex interval games. J Appl Math Decis Sci. DOI: 10.1155/2009/342089
Alparslan Gök SZ, Branzei R, Tijs S (2010) The interval Shapley value: an axiomatization. CEJOR 18:131–140
Branzei R, Alparslan Gök SZ, Branzei O (2011) Cooperation games under interval uncertainty: on the convexity of the interval undominated cores. CEJOR 19:523–532
Branzei R, Branzei O, Alparslan Gök SZ, Tijs S (2010) Cooperative interval games: a survey. CEJOR 18:397–411
Branzei R, Dimitrov D, Tijs S (2003) Shapley-like values for interval bankruptcy games. Econ Bull 3:1–8
Deng, X., Jiang, W., Zhang, J. (2017). Zero-sum matrix game with payoffs of Dempster-Shafer belief structures and its applications on sensors. Sensors, 17(4), 922.
Han W-B, Sun H, Xu G-J (2012) A new approach of cooperative interval games: the interval core and Shapley value revisited. Oper Res Lett 40:462–468
Hong, F. X., Li, D. F. (2017). Nonlinear programming method for interval-valued n-person cooperative games. Operational Research, 17(2), 479-497.
Ye, Y. F., Li, D. F. (2018). A simplified method of interval-valued solidarity values for a special class of interval-valued cooperative games. Journal of Intelligent Fuzzy Systems, (Preprint), 1-8.
Li D-F (2016) Models and Methods of Interval-Valued Cooperative Games In Economic Management. Springer, Cham
Li, D. F. (2016). Models and Methods for Interval-Valued Cooperative Games in Economic Management. Cham: Springer.
Li, D. F., Ye, Y. F. (2018). Interval-valued least square prenucleolus of interval-valued cooperative games and a simplified method. Operational Research, 18(1), 205-220.
Mallozzi L, Scalzo V, Tijs S (2011) Fuzzy interval cooperative games. Fuzzy Sets Syst 165:98–105
Mallozzi L, Scalzo V, Tijs S (2011) Fuzzy interval cooperative games. Fuzzy Sets Syst 165:98–105
Mares M (2001) Fuzzy Cooperative Games. Springer, Berlin
Meng, F., Chen, X., Tan, C. (2016). Cooperative fuzzy games with interval characteristic functions. Operational Research, 16(1), 1-24.
Moore R (1979) Methods and Applications of Interval Analysis. SIAM Stud Appl Math, Philadelphia
Singh, A., Gupta, A. (2018). Matrix Games with Interval-Valued 2-Tuple Linguistic Information. Games, 9(3), 62.
The missing part of the paper is conclusions and recommendations who and when should apply this model, and when must avoid using this model.
Strengths, weaknesses, opportunities and threats of the proposed model should be listed and highlighted at the end of the paper.
Author Response
1. The Introduction section has been extended to provide more motivation for considering the
interval-valued cooperative games.
2. The literature review has been extended by including the existing papers suggested by the reviewer.
3. A Conclusion section has been added in this revised version.
Reviewer 2 Report
The author studies the cooperative game endowed with interval-valued payoff. Specifically, the author proposes the interval-valued cores and interval-valued dominance cores based on the interval-valued payoff and the different types of orderings. The latter is a great novelty of the paper, as it differentiates it to the vast majority of the existing literature. The topic of the paper is very interesting, and it fits well the scope of the journal. Moreover, the paper is well-written and easy to follow. The author has well-thought-out the main concepts, structured well the paper with multiple theorems and propositions, providing the ability to the reader to navigate through the various concepts. However, the author should consider carefully the following comments to improve the quality and the presentation of their paper.
1. My main question throughout the whole paper is the applicability of the proposed theory. Moreover, the author has provided a pure literature review, starting the discussion only from the cooperative games. However, why the author has focused only on that type of games? What makes those game those important compared to the non-cooperative games? Also, what are the application areas that were studied by the author to get motivated about the proposed theory, e.g., Tsiropoulou, E.E., Vamvakas, P. and Papavassiliou, S., 2017. Joint customized price and power control for energy-efficient multi-service wireless networks via S-modular theory. IEEE Transactions on Green Communications and Networking, 1(1), pp.17-28, Basar, T., & Olsder, G. J. (1999). Dynamic noncooperative game theory (Vol. 23). Siam. The author is encouraged to revise the provided literature review, better motivate its research work, update the references list and give some applications examples of the proposed theory or from which applications and problems was the proposed theory motivated by.
2. Overall, the reviewer could easily follow the proposed analysis, the mathematical concepts and proofs were concrete and correct, except for the propositions 7.1 and 7.2, where the proof is not straightforwardly derived. It is not clear what the author means by “the individual rationality is not considered”. The authors should better explain the proofs of those two propositions.
3. In the whole manuscript, the author provides only toy-examples and not real applications of the proposed theory, It would be very interesting if the author could provide an additional section discussing the different applications areas of the novel theory that is introduced.
4. Also, the author should include some concluding remarks summarizing the proposed research and provide some future directions.
Minor: the author should check the whole manuscript for typos, grammar and syntax errors. The reviewer has found several typos while reading the manuscript.
Overall, this is a very interesting paper, well-aligned with the scope of the journal, with concrete and correct mathematical analysis, well-thought-out concepts and great potential of applying the proposed framework in real-life scenarios. The latter should be well-explained.
Author Response
1. The literature review has been extended in this revised version. Also, the motivation for considering this type of cooperative has been
addressed in this revised version.
2. More explanations for Propositions 7.1 and 7.2 regarding the individual rationality are provided.
3. The author appreciates the reviewer suggesting to study the different application areas of the proposed method. Since the author just owns the mathematical background, it seems that providing an additional section discussing the applications is not easy task and cannot be finished in a short time. The applications can be taken to be the future research. In other words, the author may need to find some researchers with economical background. There are six examples in this paper. Although the examples provided in this paper are not the practical problems, these examples demonstrate the possible application for the general practical problems and presents the essential ideas of the proposed method.
4. A Conclusion section has been added in this revised version.
5. The typos, grammar and syntax errors of the whole manuscript have been carefully checked again.
Reviewer 3 Report
I recommend the paper for publication. The paper is very technical but all the notions are explained so it is easy to follow the manuscript. I would like to one of two examples more.
Is Section 2 based on other work. If the answer is yes, please put some textbooks in the reference.
Author Response
The textbook for interval analysis has been added in the reference.
There are six examples in this paper.
The Introduction section has been extended to provide more motivation for considering the
interval-valued cooperative games. Also, the literature review has been extended.
A Conclusion section has been added in this revised version.
Round 2
Reviewer 1 Report
References should be numbered in order of appearance and indicated by a numeral or numerals in square brackets, e.g., [1] or [2,3], or [4–6]. References must be numbered in order of appearance in the text (including citations in tables and legends) and listed individually at the end of the manuscript.
Style of references list must strictly meet the requirements presented in the template provided by the Journal: https://www.mdpi.com/files/word-templates/mathematics-template.dot
Moreover, authors must carefully edit the references list.
Author 1, A.B.; Author 2, C.D. Title of the article. Abbreviated Journal Name Year, Volume, page range, DOI.
All equations must be numbered.
Conclusions must highlight the novelty of the proposed method and novel results. Authors must strengthen the findings.